# GraB: Finding Provably Better Data Permutations than Random Reshuffling

**Yucheng Lu, Wentao Guo, Christopher De Sa**
Department of Computer Science
Cornell University
{yl2967, wg247, cmd353}@cornell.edu

## Abstract

Random reshuffling, which randomly permutes the dataset each epoch, is widely adopted in model training because it yields faster convergence than with-replacement sampling. Recent studies indicate greedily chosen data orderings can further speed up convergence empirically, at the cost of using more computation and memory. However, greedy ordering lacks theoretical justification and has limited utility due to its non-trivial memory and computation overhead. In this paper, we first formulate an example-ordering framework named *herding* and answer affirmatively that SGD with herding converges at the rate $O(T^{-2/3})$ on smooth, non-convex objectives, faster than the $O(n^{1/3}T^{-2/3})$ obtained by random reshuffling, where $n$ denotes the number of data points and $T$ denotes the total number of iterations. To reduce the memory overhead, we leverage discrepancy minimization theory to propose an online Gradient Balancing algorithm (GraB) that enjoys the same rate as herding, while reducing the memory usage from $O(nd)$ to just $O(d)$ and computation from $O(n^2)$ to $O(n)$, where $d$ denotes the model dimension. We show empirically[1] on applications including MNIST, CIFAR10, WikiText and GLUE that GraB can outperform random reshuffling in terms of both training and validation performance, and even outperform state-of-the-art greedy ordering while reducing memory usage over $100\times$.

## 1 Introduction

Many machine learning problems can be formulated as minimizing a differentiable (loss) function $f : \mathbb{R}^d \to \mathbb{R}$ for a set of data examples $\{\boldsymbol{x}_i\}_{i=1}^n$. To train a parameterized model, the goal is to obtain a set of target parameters $\boldsymbol{w}^* = \arg\min f(\boldsymbol{w})$, where $f(\boldsymbol{w}) = \frac{1}{n}\sum_{i=1}^n f(\boldsymbol{w}; \boldsymbol{x}_i)$, and $f(\boldsymbol{w}; \boldsymbol{x}_i)$ denotes the loss incurred on the $i$-th data example $\boldsymbol{x}_i$ (usually a mini-batch of images, sentences, etc.) with model parameters $\boldsymbol{w}$. A typical model training, or optimization process, is to iteratively update the model parameter $\boldsymbol{w}$ starting from some initial $\boldsymbol{w}^{(1)}$ by running

$$\boldsymbol{w}^{(t+1)} = \boldsymbol{w}^{(t)} - \alpha\nabla f(\boldsymbol{w}^{(t)}; \boldsymbol{x}_{\sigma(t)}) \qquad t = 1, 2, \cdots \tag{1}$$

where $\alpha$ denotes the step size, and $\sigma : \{1, \ldots, n\} \to \{1, \ldots, n\}$ denotes a permutation (ordering) from which the examples are chosen to compute the stochastic gradients[2]. A widely adopted ordering protocol is Random Reshuffling (RR), where the optimizer scans in an order drawn at random without replacement over the entire training dataset multiple times during training (a common name for a single such scan is an "epoch"). RR allows the optimizer to converge faster empirically and enjoys a

---

[1]The experimental code is available at https://github.com/EugeneLYC/GraB.

[2]In this paper, we consistently use the term *stochastic gradients* to refer to gradients computed on a single (or a set of) data example(s), even when those examples are not selected at random.

36th Conference on Neural Information Processing Systems (NeurIPS 2022).

better convergence rate in theory [1]. Despite RR's theoretical improvement, it has been proven that RR does not always guarantee a good ordering [2, 3]; in fact a random permutation is far from being optimal even when optimizing a simple quadratic objective [4]. In light of this, a natural research question is:

*Can we find provably better orderings than Random Reshuffling—and do so efficiently?*

Recent studies indicate the possibility of greedily constructing better permutations using stale estimates of each $\nabla f(\boldsymbol{w}; \boldsymbol{x}_j)$ from the previous epoch [5, 6]. Concretely, Lu et al. [5] proved that for any model parameters $\boldsymbol{w} \in \mathbb{R}^d$, if sums of consecutive stochastic gradients converge faster to the full gradient, then the optimizer will converge faster. Formally, in one epoch, any permutation $\sigma$ minimizing the term (named *average gradient error*)

$$\max_{k \in \{1, \dots, n\}} \left\| \sum_{t=1}^{k} \left( \underbrace{\nabla f(\boldsymbol{w}; \boldsymbol{x}_{\sigma(t)}) - \nabla f(\boldsymbol{w})}_{\text{gradient error}} \right) \right\|, \tag{2}$$

leads to fast convergence. Leveraging this insight, Lu et al. [5] proposes to greedily select $\sigma(t)$ one at a time for all $t \in \{1, \dots, n\}$ at the beginning of the $(k + 1)$-th epoch, using the stochastic gradients computed *during* the $k$-th epoch as an estimate. This strategy works well empirically, but it remains an open question how its convergence rate compares to that of RR. Moreover, the greedy ordering method intensively consumes $O(nd)$ memory to store the gradients and $O(n^2)$ computation to order, which significantly hinders its usefulness in practice. For instance, training a simple logistic regression on MNIST would easily cost more than 1 GB additional memory compared to RR.

In this paper, we address the limitations of previous order-selection approaches, culminating in proposing a new algorithm, GraB, which converges faster than RR both in theory and in practice without any blow-up in memory or computational time. First, we address the theoretical gap by connecting the convergence of permuted-order SGD to the classic *herding* problem [7]: we show how any algorithm for solving the herding problem can be run on stale stochastic gradients to select an example ordering. While the greedy selection method of previous work may fail to select a good ordering, using the best herding algorithms from the literature yields a better convergence rate: $O(T^{-2/3})$ compared to RR's $O(n^{1/3}T^{-2/3})$ on smooth non-convex problems and $\tilde{O}(T^{-2})$ compared to RR's $\tilde{O}(nT^{-2})$ under PL condition, where $T$ is the total number of iterations. Building on this, we leverage *discrepancy minimization* theory [8] to propose an online Gradient Balancing algorithm (GraB) that achieves the same $O(T^{-2/3})$ rate but only requires $O(d)$ memory and $O(n)$ computation. Perhaps surprisingly, on multiple deep learning applications we observe GraB not only allows fast minimization of the empirical risk, but also lets the model generalize better. Our contributions in this paper can be summarized as follows:

- We formulate a general framework of sorting stochastic gradients with herding, and prove SGD with herding converges at a faster rate than Random Reshuffling (Section 3).
- We propose an online Gradient Balancing (GraB) algorithm that enjoys the same fast convergence rate as herding SGD while significantly reducing the memory from $O(nd)$ to $O(d)$ and computation from $O(n^2)$ to $O(n)$ (Section 5).
- We conduct extensive experiments on MNIST, CIFAR10, WikiText and GLUE with different models. We demonstrate that GraB converges faster in terms of both training and validation compared to Random Reshuffling and other baselines (Section 6).

## 2 Related Work

While traditional data ordering research mostly focuses on with-replacement samplers [9–11], without-replacement sampling is more common in practice [12]. Random Reshuffling (RR) [13] and the related shuffle-once (SO) method [14, 15] are among the most popular data permutation methods. Recht and Ré [16] undertakes the first theoretical investigation of RR, while subsequent works like [2, 17] give counter examples where RR orders badly. HaoChen and Sra [18], Gürbüzbalaban et al. [19], and Mishchenko et al. [1] discuss extensively on the conditions needed for RR to benefit. Some recent works [5, 6] suggest constructing better data permutations than RR via a memory-intensive greedy strategy. Concretely, Mohtashami et al. [6] proposes evaluating gradients on all the examples first to minimize Equation (2) before starting an epoch, applied to Federated Learning; Lu et al. [5]

**Algorithm 1** Herding with Greedy Ordering [5]

1: **Input:** a group of vectors $\{z_i\}_{i=1}^n$.
2: Center all the vectors: $z_i \leftarrow z_i - \frac{1}{n} \sum_{j=1}^n z_j, \forall i \in [n]$.
3: Initialize an arbitrary $\sigma$, running partial sum: $s \leftarrow \mathbf{0}$, candidate set $\Phi \leftarrow \{1, \cdots, n\}$.
4: **for** $i = 1, \ldots, n$ **do**
5:     Iterate through $\Phi$ and select $z_j$ from $\Phi$ that minimizes $\|s + z_j\|$.
6:     Remove $j$ from $\Phi$, update partial sum and order: $s \leftarrow s + z_j; \sigma(i) \leftarrow j$.
7: **end for**
8: **return** $\sigma$.

---

**Algorithm 2** General Framework of SGD with Offline Herding

1: **Input:** number of epochs $K$, initialized order $\sigma_1$, initialized weight $w_1$, learning rate $\alpha$.
2: **for** $k = 1, \ldots, K$ **do**
3:     **for** $t = 1, \ldots, n$ **do**
4:         Compute gradient $\nabla f(w_k^{(t)}; x_{\sigma_k(t)})$.
5:         Store the gradient: $z_t \leftarrow \nabla f(w_k^{(t)}; x_{\sigma_k(t)})$.
6:         Optimizer Step: $w_k^{(t+1)} \leftarrow w_k^{(t)} - \alpha \nabla f(w_k^{(t)}; x_{\sigma_k(t)})$.
7:     **end for**
8:     Generate new order: $\sigma_{k+1} \leftarrow \texttt{Herding}(\{z_t\}_{t=1}^n)$.
9:     $w_{k+1}^{(1)} \leftarrow w_k^{(n+1)}$.
10: **end for**
11: **return** $w_K^{(1)}$.

---

provides an alternative of minimizing Equation (2) using stale gradients from previous epoch to estimate the gradient on each example. Rajput et al. [4] introduces an interesting variant to RR by reversing the ordering every other epoch, achieving better rates for quadratics. Other approaches, such as curriculum learning [20], try to order the data to mimic human learning and improve generalization [21–23]: these approaches differ from our goal of finding good permutations for minimizing loss in a finite-sum setting.

## 3 Offline Stale-Gradient Herding

In this section, we study the use of *stale gradients*—the stochastic gradients for each example from the previous epoch, saved in memory—to construct a data permutation at the start of each epoch. Lu et al. [5] and Mohtashami et al. [6] propose to use greedy ordering in this way—selecting examples one at a time to minimize Equation (2). Here, we first show that this objective is closely related to the classic *herding* problem from discrepancy theory. We demonstrate with examples from the herding formulation that in the adversarial setting, greedy selection is bad at herding and can underperform basic random reshuffling. We conclude this section by proving SGD with proper herding converges faster than random reshuffling.

**Herding.** The *herding* problem originates from Welling [24] for sampling from a Markov random field that agrees with a given data set. Its discrete version is later formulated in Harvey and Samadi [7]. Given $n$ vectors $\{z_i\}_{i=1}^n$ that are $d$-dimensional (i.e. in $\mathbb{R}^d$) and have norm $\|z_i\|_2 \le 1$, the herding problem we study[3] is the task of finding a permutation $\sigma^* : [n] \to [n]$ that minimizes

$$\max_{k \in \{1,\ldots,n\}} \left\| \sum_{t=1}^k \left( \underbrace{z_{\sigma(t)} - \frac{1}{n} \sum_{i=1}^n z_i}_{\text{vector error}} \right) \right\|_\infty. \tag{3}$$

State-of-the-art algorithms guarantee an $\tilde{O}(1)$ bound to this objective [7]. It is straightforward to observe the the formulation of Equation (3) generalizes Equation (2) if we replace each $z_i$ with the stochastic gradient computed on the example $i$. In other words, any algorithm that minimizes Equation (3) can be used to minimize Equation (2) and find a better data permutation. The simplest way to do this is with the greedy algorithm for herding, which we show in Algorithm 1: this is

---

[3]A small distinction is Harvey and Samadi [7] investigates infinite sequences while here we are studying a finite ordering, i.e. a permutation.

Table 1: Table summarizing the theoretical results in this paper, where $n$ denotes the total number of data points (examples), $d$ denotes the model dimension and $T$ denotes the total number of iterations ($n$ times the number of epochs).

| | Rate (Non-convex) | Rate (PL) | Computation over RR | Storage over RR |
|---|---|---|---|---|
| RR | $O(n^{\frac{1}{3}}T^{-\frac{2}{3}})$ | $O(nT^{-2})$ | N/A | N/A |
| Herding | $\tilde{O}(T^{-\frac{2}{3}})$ | $\tilde{O}(T^{-2})$ | $O(n^2)$ | $O(nd)$ |
| GraB | $\tilde{O}(T^{-\frac{2}{3}})$ | $\tilde{O}(T^{-2})$ | $O(n)$ | $O(d)$ |

essentially what the previous work was doing to select an order based on stale gradients. Unfortunately, despite its elegance, it was first pointed out in Chelidze et al. [25] that the greedy ordering can underperform random permutation. We adapt their result to show the following.

**Statement 1.** *There exist $n$ vectors in $\mathbb{R}^2$ such that when applying Algorithm 1, the objective in Equation (2) becomes $\Omega(n)$; a random permutation is guaranteed to achieve $O(\sqrt{n})$ on average.*

**Herding SGD with Stale Gradients.** Based on the herding framework, a path to SGD with better data permutation becomes clear: when training models during a epoch, we can simply store all the stochastic gradients, and then herd them (with some herding algorithm) offline at the start of the next epoch to obtain the order for that epoch. This stale-gradient approach is formally described in Algorithm 2, and could be contrasted with a fresh-gradient approach (as in Mohtashami et al. [6]) that herded with newly computed stochastic gradients at the start of each epoch—it is desirable to avoid the use of fresh gradients if possible as they double the gradient computations needed each epoch. The greedy ordering of Lu et al. [5] can be described as running Algorithm 2 using Algorithm 1 as the `Herding` subroutine. However, as greedy herding with Algorithm 1 does not have a good worst-case guarantee, we propose using some other herding algorithm instead. There have been many algorithms proposed that can reduce the Equation (3) to $\tilde{O}(1)$ in polynomial time. Here, we show that this herding-objective bound is sufficient to prove convergence of herded SGD and even yield a better convergence rate than Random Reshuffling. We defer the details of the herding algorithm to Section 5, as they do not affect the convergence rate of SGD.

We start by stating some assumptions for non-convex optimization, as well as our herding bound.

**Assumption 1.** *(**Smoothness**.) For any example $x_j$ in the dataset (for $j \in \{1, \ldots, n\}$), the loss for $x_j$ is $L_{2,\infty}$-smooth and $L_\infty$-smooth meaning that, for any $w, v \in \mathbb{R}^d$, it holds that*

$$\|\nabla f(w; x_j) - \nabla f(v; x_j)\|_2 \leq L_{2,\infty}\|w - v\|_\infty \qquad and$$
$$\|\nabla f(w; x_j) - \nabla f(v; x_j)\|_\infty \leq L_\infty\|w - v\|_\infty.$$

*The averaged loss function is also $L$-smooth in that, for any $w, v \in \mathbb{R}^d$, it holds that*

$$\|\nabla f(w) - \nabla f(v)\|_2 \leq L\|w - v\|_2.$$

**Assumption 2.** *(**PL condition**.) We say the loss function $f$ fulfills the Polyak-Lojasiewicz (PL) condition if there exists $\mu > 0$ such that for any $w \in \mathbb{R}^d$,*

$$\frac{1}{2}\|\nabla f(w)\|_2^2 \geq \mu\left(f(w) - f^*\right), \qquad where \quad f^* = \inf_{v \in \mathbb{R}^d} f(v).$$

**Assumption 3.** *(**Bounded Gradient Error**.) For any $j \in \{1, \cdots, n\}$ and any $w \in \mathbb{R}^d$,*

$$\left\|\nabla f(w; x_j) - \tfrac{1}{n}\sum_{s=1}^n \nabla f(w; x_s)\right\|_2 \leq \varsigma.$$

**Assumption 4.** *(**Herding Bound**.) In Algorithm 2, the subroutine `Herding` algorithm has the following property [26]: if given input vectors $z_1, \ldots, z_n \in \mathbb{R}^d$ that satisfy $\|z_i\|_2 \leq 1$ and $\sum_{i=1}^n z_i = 0$, `Herding` outputs a permutation $\sigma$ of $\{1, \ldots, n\}$ such that for all $k \in \{1, \ldots, n\}$, $\|\sum_{i=1}^k z_{\sigma(i)}\|_\infty \leq H$ for some constant $H$.*

**Remarks on the assumptions.** Assumptions 1, 2, and 3 are commonly used assumptions in the study of permutation-based SGD, except that we separate different smoothness constants with respect to different norms to tighten the final bound. The cross-norm Lipschitz constant $L_{2,\infty}$ is often used to obtain tight dimension-independent bounds [27, 28]. Note that these cross-norm assumptions can

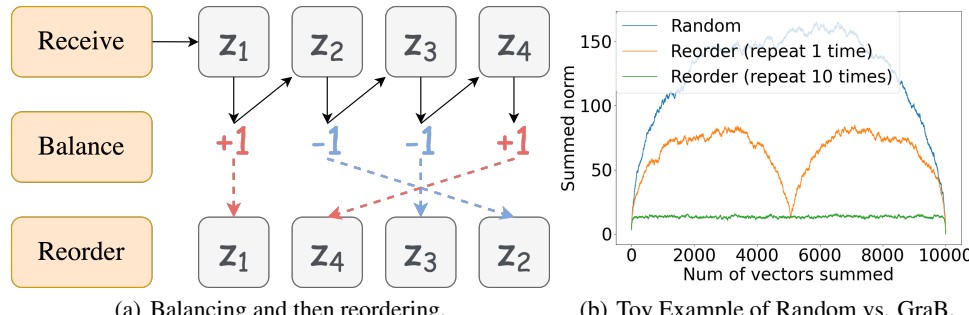

| | |
|---|---|
| (a) Balancing and then reordering. | (b) Toy Example of Random vs. GraB. |

Figure 1: (a) illustrates how Algorithm 3 reorders the vectors with balancing – the new order is obtained by concatenating *original* order of the examples with $+1$, followed by the *reverse* order of the examples with $-1$; (b) demonstrates a toy example on $n = 10000$ vectors $\{z_i\}_{i=1}^n$ where $z_i$ is randomly sampled from $[0, 1]^{128}$. For a given order $\sigma$, we plot the norm of the prefix sums $\|\sum_{t=1}^k \left( z_{\sigma(t)} - 1/n \sum_{s=1}^n z_s \right)\|$ for all $k = 1, \ldots, n$.

be dropped by incurring extra factors of the dimension $d$, as $L_\infty \leq L_{2,\infty} \leq \sqrt{d}L$ for any function. Assumption 4 holds for multiple algorithms for some $H = \tilde{O}(1)$: we will discuss this in depth in Section 5, while treating $H$ as a parameter here as we want to observe the dependence on this variable in the final bound. Under these assumptions, the convergence rate of Algorithm 2 is given in the following theorem.

**Theorem 1.** *In Algorithm 2, under Assumptions 1, 3 and 4, if we set $\alpha$ to be*

$$\alpha = \min \left\{ \sqrt[3]{\frac{f(w_1) - f^*}{24nH^2\varsigma^2 L_{2,\infty}^2 K}}, \frac{1}{8n(L + L_{2,\infty})}, \frac{1}{32nL_\infty}, \frac{1}{16HL_{2,\infty}} \right\}$$

*then it converges at the rate*

$$\frac{1}{K} \sum_{k=1}^K \|\nabla f(w_k)\|^2 \leq 36 \sqrt[3]{\frac{H^2\varsigma^2 L_{2,\infty}^2 F^2}{n^2 K^2}} + \frac{\varsigma^2}{K} + \frac{32F(L + L_{2,\infty} + L_\infty)}{K} + \frac{64FHL_{2,\infty}}{nK},$$

*where $F = f(w_1) - f^*$. Furthermore, under the additional PL condition (Assumption 2), with $\alpha$ as*

$$\alpha = \min \left\{ \frac{1}{n\mu}, \frac{1}{48HL_{2,\infty}}, \frac{1}{96n(L + L_{2,\infty} + L_\infty)}, \frac{2}{n\mu K} W_0 \left( \frac{(f(w_1) - f^* + \varsigma^2)n^2\mu^3 K^2}{192H^2 L_{2,\infty}^2 \varsigma^2} \right) \right\},$$

*where $W_0$ denotes the Lambert W function, Algorithm 2 converges at the rate*

$$f(w_K) - f^* \leq \tilde{O} \left( \frac{H^2 L_{2,\infty}^2 \varsigma^2}{\mu^3 n^2 K^2} \right).$$

**Improved Rate Compared to RR.** Theorem 1 gives us a better rate than random reshuffling, in that there is no dependence on the number of examples $n$ in the dominant term. If we let $T = nK$ be the total number of steps, then Theorem 1 shows Algorithm 2 converges at rate $O(\frac{1}{T^{2/3}} + \frac{n}{T})$. In contrast, random reshuffling gives a rate of $O(\frac{n^{1/3}}{T^{2/3}} + \frac{n}{T})$ as given in Mishchenko et al. [1], Theorem 4. Similarly, herding improves the rate from $\tilde{O}(\frac{n}{T^2})$ to $\tilde{O}(\frac{1}{T^2})$ in the PL case. This gives us an affirmative answer to our question from the intro: a faster rate *can* be obtained even if we herd with stale gradients estimates, without computing additional gradients.

## 4 Enabling Memory-Efficient Herding with Balancing

In Section 3 we introduce the herding problem, and prove that with proper herding, SGD converges at a faster rate than Random Reshuffling. However, a downside of offline herding SGD is that it

---

**Algorithm 3** Reordering Vectors based on Balanced Signs [7]

---

1: **Input:** a group of signs $\{\epsilon_i\}_{i=1}^n$ for $\{z_i\}_{i=1}^n$ from a balancing algorithm, initial order $\sigma$
2: Initialize two order-sensitive lists $L_{\text{positive}} \leftarrow [\,], L_{\text{negative}} \leftarrow [\,]$.
3: **for** $i = 1, \ldots, n$ **do**
4:     Append $\sigma(i)$ to $L_{\text{positive}}$ if $\epsilon_i$ is $+1$ else append it to $L_{\text{negative}}$.
5: **end for**
6: **return** new order $\sigma' = \text{concatenate}(L_{\text{positive}}, \text{reverse}(L_{\text{negative}}))$.

---

requires storing all the vectors (i.e., in the case of SGD, all the stochastic gradients) which incurs $O(nd)$ memory overhead. In this section, we present background on the *vector balancing problem*, the solutions of which will enable us to solve the herding problem in an online fashion.

**Background: Online Vector Balancing.** The problem of *online vector balancing* was first formulated in Spencer [29]: given $n$ vectors $\{z_i\}_{i=1}^n \in \mathbb{R}^d$, arriving one at a time, the goal is to assign a sign $\epsilon_i \in \{-1, +1\}$ to each vector upon receiving it so as to minimize $\max_{k \in [n]} \| \sum_{i=1}^k \epsilon_i z_i \|_\infty$. Over the last few decades, many efforts have been spent on bounding this $\ell_\infty$-norm objective. Specifically, Aru et al. [30] and Bansal and Spencer [31] shows it can be made to be on the order of $\tilde{O}(1)$ when the coordinates of each vector are *independent*. Bansal et al. [32] gives a $\texttt{poly}(\log(n), d)$ bound when there are dependencies among coordinates. The adversarial setting with interval discrepancy is discussed in [33]. Other works focus on balancing variants such as stochastic arrival [34] or box discrepancy [35].

**Solving Herding via Balancing.** While the objectives of balancing and herding are different, an algorithm for balancing can be used as a subroutine to do herding as well. Consider the simple case of vectors $z_1, \ldots, z_n$ in some order, where $\sum_{i=1}^n z_i = 0$, and suppose we have obtained a group of signs $\{\epsilon_i\}_{i=1}^n$ via some balancing algorithm: can we use this to determine an order of $\{z_i\}_{i=1}^n$ that can solve the herding problem? Harvey and Samadi [7], Chobanyan et al. [36], and Chobanyan [37] approach this problem by a clever *reordering* that works as in Algorithm 3. Concretely, a new order is obtained by first taking all the vectors with positive signs in order, followed by all the vectors with negative signs in reversed order from the original sequence. This process is visualized in Figure 1. Harvey and Samadi [7] shows that this balancing-and-then-reordering has the following effect on the herding objective in Equation (3).

**Theorem 2.** *([7], Theorem 10) Consider $n$ vectors $z_1, \ldots, z_n \in \mathbb{R}^d$ that fulfill $\sum_{i=1}^n z_i = 0$ and $\|z_i\| \leq 1$. Suppose that for some permutation $\sigma$, the herding objective is bounded by $\max_k \| \sum_{i=1}^k z_{\sigma(i)} \|_\infty \leq H$ for some constant $H$. Suppose we have computed some signs $\epsilon_1, \ldots, \epsilon_n$ such that the vector-balancing objective $\max_k \| \sum_{i=1}^k \epsilon_i z_{\sigma(i)} \|_\infty \leq A$ for some constant $A$. Then if we let $\sigma'$ be the permutation output by Algorithm 3, the herding objective on this new permutation is bounded by $\max_k \| \sum_{i=1}^k z_{\sigma'(i)} \|_\infty \leq (A + H)/2$.*

Theorem 2 shows that calling Algorithm 3 allows us to construct a new ordering from balancing, which reduces the current herding bound by a constant factor if $A \leq H$. This means that if we have a balancing algorithm that is guaranteed to reduce the balancing objective to less than $A$, by applying this "reordering" many times, we can also reduce the herding objective to be less than $A$: this approach is used in many offline herding algorithms that achieve $\tilde{O}(1)$ bounds.

## 5 GraB: SGD with Gradient Balancing for Good Data Permutation

Building on the balance-then-reorder paradigm from the previous section, in this section we propose an algorithm named GraB that improves SGD with herding by choosing the ordering *online*—without storing gradients—thus using only $O(d)$ memory and $O(n)$ computation. However, the conditions of applying the balancing-reordering framework bring up additional challenges in this online setting:

- **Challenge I.** To apply Theorem 2, we will need to pre-center all the vectors (as in line 2 of Algorithm 1) to ensure they sum to zero. This cannot be done online, as we need to make decisions on-the-fly before seeing all the vectors.
- **Challenge II.** The reordering strategy, as shown in Theorem 2, only reduces the herding bound by a constant factor, which does not give us the accelerated rate directly since applying Theorem 1

---

**Algorithm 4** SGD with Online **Gra**dient **B**alancing (**GraB**)

---
1: **Input:** number of epochs $K$, initialized order $\sigma_1$, initialized weight $\boldsymbol{w}_1$, stale mean $\boldsymbol{m}_1 = \boldsymbol{0}$, step size $\alpha$.
2: **for** $k = 1, \ldots, K$ **do**
3:     Initialized indices and running average: $l \leftarrow 1$; $r \leftarrow n$; $\boldsymbol{s} \leftarrow \boldsymbol{0}$; $\boldsymbol{m}_{k+1} \leftarrow \boldsymbol{0}$.
4:     **for** $t = 1, \ldots, n$ **do**
5:         Compute gradient $\nabla f(\boldsymbol{w}_k^t; \boldsymbol{x}_{\sigma_k(t)})$.
6:         Compute centered gradient and update stale mean:
$$\boldsymbol{g}_{k,t} \leftarrow \nabla f(\boldsymbol{w}_k^t; \boldsymbol{x}_{\sigma_k(t)}) - \boldsymbol{m}_k$$
$$\boldsymbol{m}_{k+1} \leftarrow \boldsymbol{m}_{k+1} + \nabla f(\boldsymbol{w}_k^t; \boldsymbol{x}_{\sigma_k(t)})/n$$
7:         Compute sign for the current gradient: $\epsilon_{k,t} \leftarrow \texttt{Balancing}(\boldsymbol{s}, \boldsymbol{g}_{k,t})$
8:         **if** $\epsilon_{k,t} = +1$ **then**
9:             $\boldsymbol{s} \leftarrow \boldsymbol{s} + \boldsymbol{g}_{k,t}$; $\sigma_{k+1}(l) \leftarrow \sigma_k(t)$; $l \leftarrow l + 1$.
10:         **else**
11:             $\boldsymbol{s} \leftarrow \boldsymbol{s} - \boldsymbol{g}_{k,t}$; $\sigma_{k+1}(r) \leftarrow \sigma_k(t)$; $r \leftarrow r - 1$.
12:         **end if**
13:         Optimizer Step: $\boldsymbol{w}_k^{(t+1)} \leftarrow \boldsymbol{w}_k^{(t)} - \alpha \nabla f(\boldsymbol{w}_k^t; \boldsymbol{x}_{\sigma_k(t)})$.
14:     **end for**
15:     $\boldsymbol{w}_{k+1}^{(1)} \leftarrow \boldsymbol{w}_k^{(n+1)}$.
16: **end for**
17: **return** $\boldsymbol{w}_K^{(1)}$.

---

---

**Algorithm 5** Balancing without normalization.

---
1: **Input:** current sum $\boldsymbol{s}$, vector $\boldsymbol{v}$.
2: $\epsilon \leftarrow +1$ **if** $\|\boldsymbol{s} + \boldsymbol{v}\| < \|\boldsymbol{s} - \boldsymbol{v}\|$ **else** $\epsilon \leftarrow -1$.
3: **return** $\epsilon$.

---

    would require Equation (3) to be on the order of $\tilde{O}(1)$. We can repeatedly call the balancing-reordering subroutines, but that would again require storing all the stochastic gradients.

To overcome challenge I, we apply a two-step stale gradient estimate: for any $k$, we use the running average of stale gradients to "center" stochastic gradient in epoch $k + 1$ (this centering is itself stale, and does not guarantee the vectors sum to $0$, but this is still enough to prove convergence). Then, the online balancing-reordering subroutine will determine the proper order to use in epoch $k + 2$. To address challenge II, we leverage a noisy reordering process: let $\sigma_k$ denote the order to use in epoch $k$, then the next order $\sigma_{k+1}$ will be obtained from $\sigma_k$. The intuition here is that if we train the model with enough epochs, the reordering will in general push the herding bound towards $A$ in Theorem 2 without storing any additional vectors. We present the full GraB algorithm in Algorithm 4.

We proceed to provide a convergence guarantee for Algorithm 4. Since now we are working with balancing rather than herding, in the convergence analysis we replace Assumption 4 with the following one related to balancing.

**Assumption 5.** (*Balancing Bound.*) *There exists a constant $A > 0$ such that in Algorithm 4, if the subroutine* `Balancing` *is given input vectors $\boldsymbol{z}_1, \ldots, \boldsymbol{z}_n \in \mathbb{R}^d$ that satisfy $\|\boldsymbol{z}_i\|_2 \leq 1$, then* `Balancing` *outputs signs $\epsilon_i \in \{-1, +1\}$ such that $\|\sum_{i=1}^k \epsilon_i \boldsymbol{z}_{\sigma(i)}\|_\infty \leq A$ for all $k \in \{1, \ldots, n\}$.*

We defer discussing the complexity of $A$ to the later part of this section, and first present the convergence guarantee for GraB (Algorithm 4) as follows.

**Theorem 3.** *In Algorithm 4, under Assumptions 1, 3 and 5, if we set $\alpha$ to be*

$$\alpha = \min\left\{ \sqrt[3]{\frac{f(\boldsymbol{w}_1) - f^*}{32nA^2\varsigma^2 L_{2,\infty}^2 K}}, \frac{1}{nL}, \frac{1}{26(n+A)L_{2,\infty}}, \frac{1}{260nL_\infty} \right\}$$

*then it converges at the rate*

$$\frac{1}{K}\sum_{k=1}^K \|\nabla f(\boldsymbol{w}_k)\|^2 \leq 11\sqrt[3]{\frac{H^2\varsigma^2 L_{2,\infty}^2 F^2}{n^2 K^2}} + \frac{\varsigma^2}{K} + \frac{65(F(L + L_{2,\infty} + L_\infty)}{K} + \frac{8FAL_{2,\infty}}{nK}.$$

---
**Algorithm 6** Probabilistic Balancing with Logarithm Bound. [38]
---
1: **Input:** current sum $s$, received vector $z$, hyperparameter $c$.
2: **if** $|s, z| > c$ or $\|s\|_\infty > c$ **then**
3:     **Fail.**
4: **end if**
5: $\epsilon \leftarrow +1$ with probability $\frac{1}{2} - \frac{\langle s, z \rangle}{2c}$ and $\epsilon \leftarrow -1$ with probability $\frac{1}{2} + \frac{\langle s, z \rangle}{2c}$.
6: **return** $\epsilon$.
---

*where $F = f(\boldsymbol{w}_1) - f^*$. Furthermore, under the additional PL condition (Assumption 2) with $\alpha$ as*

$$\alpha = \min\left\{ \frac{1}{n\mu}, \frac{1}{nL}, \frac{1}{52(n+A)L_{2,\infty}}, \frac{1}{520nL_\infty}, \frac{2}{n\mu K}W_0\left( \frac{(f(\boldsymbol{w}_1) - f^* + \varsigma^2)n^2\mu^3 K^2}{256A^2 L_{2,\infty}^2 \varsigma^2} \right) \right\},$$

*where $W_0$ denotes the Lambert W function, it converges at the rate*

$$f(\boldsymbol{w}_K) - f^* \leq \tilde{O}\left( \frac{A^2 L_{2,\infty}^2 \varsigma^2}{\mu^3 n^2 K^2} \right).$$

Comparing Theorem 1 and 3, we can observe GraB (Algorithm 4) achieves essentially the same fast asymptotic convergence rate as SGD with herding (Algorithm 2).

**Asymptotic complexity for $A$ and $H$.** So far we have covered the main design of GraB. A remaining yet unanswered question is how do $A$ in Assumption 4 and $H$ in Assumption 5 relate to other factors exactly, and what are the concrete algorithms to achieve small $A$ and $H$. Some recent progress in the theory community shows that there exists an online algorithm (Algorithm 6) that guarantees with high probability that $A$ is on the order of $\tilde{O}(1)$ as shown in the following theorem.

**Theorem 4.** *(Alweiss et al. [38]) In Algorithm 6, if all the vectors have $\|\boldsymbol{z}_t\| \leq 1$ and the parameter $c$ is set $c = 30 \log(nd/\delta)$, then with probability $1 - \delta$, the maximum partial signed sum is bounded by $\max_t \|\sum_{j=1}^t \epsilon_j \boldsymbol{z}_j\|_\infty \leq c = O(\log(nd)) = \tilde{O}(1)$.*

This theorem gives us a high-probability bound on $A$ that can easily be converted to a high-probability statement that the result of Theorem 3 holds with $A$ replaced by $\tilde{O}(1)$. Theorem 4 also yields a good offline herding algorithm and a bound on $H$: recall from Theorem 2 that given any herding bound $H$ on an ordering, running Algorithm 3 once allows us to obtain a new herding bound $(A + H)/2$. Given Theorem 4 that a $\tilde{O}(1)$ bound is guaranteed on $A$, then if we repeatedly run Algorithm 3 as described in Section 4, we can eventually push the herding bound $H$ towards $A$, which yields $H = \tilde{O}(1)$. This can be done with probability 1 offline by simply restarting on failure; however, as stated in Section 3, this still requires storing all the vectors.

**On the normalization of vectors.** Note that a typical *balancing* subroutine (e.g. Algorithm 6) requires normalized vectors $\|\boldsymbol{z}_i\| \leq 1. \forall i$. This, in practice, implies we will need to estimate a large enough constant to normalize gradients when we run GraB. To alleviate this issue, we can instead adopt a balancing subroutine that is invariant to the normalizer, for instance, Algorithm 5. As shown in the next section, this naive balancing algorithm suffices for non-trivial improvements.

## 6   Experiment

In this section, we evaluate GraB and other baseline algorithms on multiple machine learning applications. We compare GraB (Algorithm 4) with the following baselines: Random Reshuffling (RR), Shuffle Once (SO) [1], FlipFlop [4] and Greedy Ordering[4] [5]. All the experiments run on an instance configured with a 4-core Intel(R) Xeon(R) 2.50GHz CPU, 32GB memory and an NVIDIA GeForce RTX 2080 Ti GPU. Although we used Algorithm 6 to show a $\tilde{O}(1)$ bound can be obtained for herding, in our experiments, we adopt the simple balancing algorithm of Algorithm 5, which performs well in practice. More results with other balancing algorithms can be found in the Appendix.

---

[4]For FlipFlop, we follow the exact implementation code from https://github.com/shashankrajput/flipflop; and for Greedy Ordering, we strictly follow https://github.com/EugeneLYC/qmc-ordering.

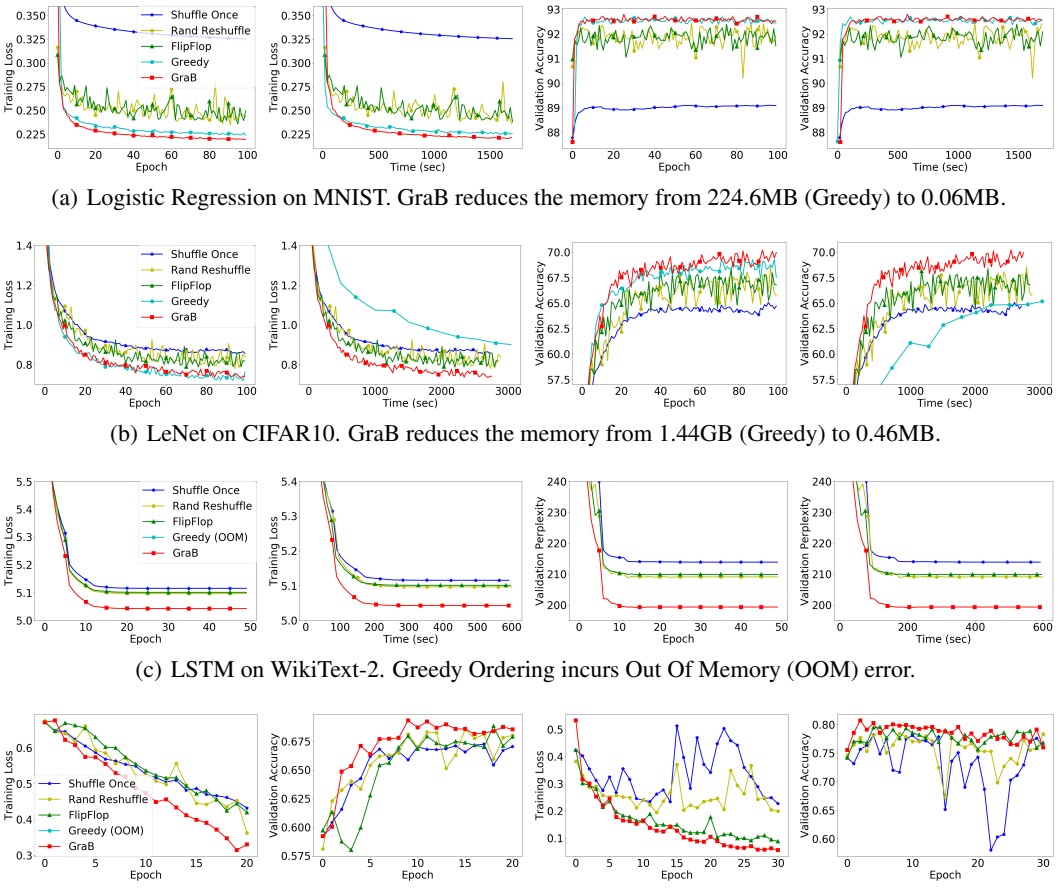

(a) Logistic Regression on MNIST. GraB reduces the memory from 224.6MB (Greedy) to 0.06MB.

(b) LeNet on CIFAR10. GraB reduces the memory from 1.44GB (Greedy) to 0.46MB.

(c) LSTM on WikiText-2. Greedy Ordering incurs Out Of Memory (OOM) error.

(d) Finetune BERT-Tiny on two GLUE tasks (left: QNLI; right: SST-2). Greedy Ordering incurs Out Of Memory (OOM) error on both tasks, and thus is not shown.

Figure 2: Training/Validation on MNIST, CIFAR10, WikiText-2 and GLUE with different models.

**Model and Dataset.** We adopt the following model training tasks for evaluation: (1) training logistic regression on MNIST, (2) training LeNet on CIFAR10 [39], (3) training LSTM on wikitext-2 [40], and (4) finetuning BERT-Tiny [41] on GLUE [42]. Detailed information regarding models, datasets and hyperparameters can be found in Appendix A.

**Convergence Speedup.** The convergence plots for each algorithms are summarized in Figure 2. We observe that in general, GraB is able to converge faster in terms of both training and validation loss on various tasks. Of all the baseline algorithms, Greedy Ordering is able to achieve comparable convergence speed with respect to epochs to GraB. However, Greedy Ordering consumes much more memory and wall-clock time as shown in the figure. In the BERT training case, we observe the Greedy Ordering runs *Out Of Memory (OOM)*, causing an error on our machine. Based on simple calculations, GraB only requires less than 1% the memory used by Greedy Ordering for all the tasks. It is worth pointing out that throughout the experiments, we do not tune hyperparameters for GraB but let it reuse the hyperparameters from RR (details in the Appendix). This implies GraB can provide in-place improvement without any additional tuning in practice, and a well-tuned GraB could benefit with even larger margin. Shuffle Once (SO) performs worst in all the cases, and FlipFlop is generally the same as RR, which is aligned with the conclusion made in Rajput et al. [4].

**Ablation Study: are good permutations fixed?** In GraB, the example ordering $\sigma$ is carried and improved over epochs, which is motivated from minimizing the herding bound. This raises another question of whether the found permutation changes at later stage of training, and if so, can we find a fixed permutation that outperforms the RR in practice? Here we conduct an ablation study on two fixed order strategies with GraB. We include the following two variants:

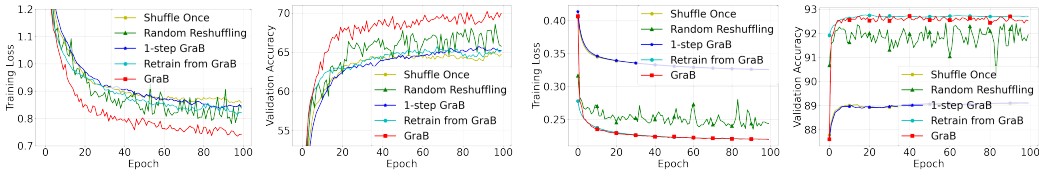

(a) LeNet/CIFAR10 (non-convex).    (b) logistic regression/MNIST (convex).

Figure 3: Ablation Study of GraB with fixed orders. *1-step GraB* refers to the algorithm that uses GraB only in the first epoch to obtain an order, and then use it as a fixed order for the rest of the training. *Retrain from GraB* refers to the algorithm that uses a fixed order obtained at the end of the epoch from a GraB full run (the latter one is not for practice, only for understanding GraB with fixed orders). The main takeaway here is that for convex problems, we can find a fixed example order that outperforms both RR and SO.

- *1-step GraB*: it only runs GraB for 1 epoch to obtain an ordering, and then use it as a fixed order for the rest of the training.
- *Retrain from GraB*: First launch a full run of GraB, and then use the ordering in the final epoch as a fixed order in a new run.

We test these new variants on two tasks: LeNet on CIFAR10 (non-convex) and logistic regression on MNIST (convex). The training loss and validation accuracy curves can be found in Figure 3. We observe that on both tasks, the *1-step GraB* does not work well, which is aligned with our theory and motivation in Section 5 Challenge II. On the other hand, *Retrain from GraB* achieves the comparable performance than original GraB on the convex task, but not on the non-convex one. This is mainly because a good ordering is going to depend on the local optimum the algorithm is approaching and for a convex problem, there is only one such optimum. It again verifies that GraB finds better orderings compared to baseline Random Reshuffling and Shuffle Once.

**On the granularity of example ordering.** Throughout this paper, we have discussed how to find good data permutations via per-example gradients. In practice, however, per-example gradients are usually not easy to obtain since data is usually loaded in batches, and many ML libraries (e.g. PyTorch) directly accumulate gradients over the batch. A direct workaround is to fix the data within batches and reorder the batches (treating them as coarse-grained examples). This, however, would compromise the benefits of ordering since the total number of examples $n$ is reduced by a factor of batch size, while the statistical improvement of herding (GraB) is in the order of $O(n^{-1/3})$. To alleviate this, we provide two alternatives: (1) Use ML frameworks that support quick per-example gradients computation (e.g. JAX[5]); (2) Leverage gradient accumulation steps as used in large language model training, i.e., use smaller batch sizes in the code but perform `optimizer.step` once every few steps so that we can obtain the finer-grained gradients on-the-fly while still optimizing the models with the desired batch size. This is the default method we used throughout the experiments, we will include more details in the appendix.

## 7    Conclusion

In this paper, we formulate a herding framework for data-ordering in SGD. We prove SGD with herding finds better data orderings, and converges faster than random reshuffling on smooth non-convex and PL objectives. We propose an online gradient balancing algorithm named GraB that finds a better ordering with little compute or memory overhead. We substantiate our theory and the usefulness of GraB on multiple machine learning applications.

## Acknowledgement

This work is supported by NSF-2046760. Yucheng is supported by Meta PhD Fellowship. The authors would like to thank Si Yi Meng and anonymous reviewers from NeurIPS 2022 for providing valuable feedbacks on this paper.

---

[5]https://jax.readthedocs.io/en/latest/jax-101/04-advanced-autodiff.html#per-example-gradients

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
