# A Experimental Details

**Setup.** We first provide the details to models and datasets used in the experiments:

1. **Logistic Regression on MNIST**: The MNIST consists of handwritten digits 0-9, it has a training set of $60,000$ examples, and a test set of $10,000$ examples. It contains $28 \times 28 = 784$ features and 10 classes. The model is of dimension $d = 784 \times 10 + 10 = 7850$ (with bias).
2. **LeNet on CIFAR10.** The CIFAR10 dataset consists of $60000$ $32 \times 32$ colour images in 10 classes, with 6000 images per class. There are 50000 training images and 10000 test images. The dataset is divided into five training batches and one test batch, each with 10000 images. The test batch contains exactly 1000 randomly-selected images from each class. LeNet is a classic convoluntional neural network proposed by [43].
3. **LSTM on WikiText-2.** In this task we train a 2-layer LSTM on the wikitext-2 dataset, which contains 2 million words. We set the embedding size to be 32, number of hidden unit to be 32 and number of head to be 2. We adopt the learning rate schedule from PyTorch example repo[6]. We set the sequence length (bptt) to be 35.
4. **BERT on GLUE.** The General Language Understanding Evaluation (GLUE) benchmark is a collection of resources for training, evaluating, and analyzing natural language understanding systems. GLUE consists of 11 different tasks. In the main paper, we evaluate on the SST-2 and QNLI. For this task, we adopt the BERT-Tiny model released by Google Research[7]. BERT-Tiny contains 768 hidden layers. For each task, we set maximum sequence length to be 32 and enable padding.

**Hyperparameters.** We tune the hyperparameters for all the algorithms except GraB within a given range. Then we reuse the hyperparameters for RR in GraB. This implies GraB can potentially provide in-place benefit without additional tuning. We use momentum SGD (with its default value 0.9) for all the tasks. The hyperparameters (ranges) for each task are as follows:

1. **MNIST.**: `LR`$\in \{0.1, 0.01, 0.001, 0.0001\}$; `BSZ=64`; `GCC=32`; `WD=0.0001`.
2. **CIFAR10.** `LR`$\in \{0.1, 0.01, 0.001, 0.0001\}$; `BSZ=16`; `GCC=2`; `WD=0.0001`.
3. **WikiText-2.** We set momentum to be 0.9 and let the learning rate follow `ReduceLROnPlateau` from Pytorch with variable: {mode='min', `factor=0.1`, `patience=5`, `threshold=5`}. The initial learning rate is set to be 5.
4. **GLUE.** On the SST-2 we adopt `BSZ=1`; `GCC=1`; `WD=0.01`; `LR`$\in \{0.005, 0.001, 0.0005, 0.0001\}$. On the QNLI task we adopt `BSZ=2`; `GCC=2`; `WD=0.01`; `LR`$\in \{0.005, 0.001, 0.0005, 0.0001\}$.

`LR` stands for learning rate, `BSZ` stands for batch size, `GCC` stands for the gradient accumulation steps and `WD` stands for the weight decay.

**Gradient Accumulation.** As illustrated in the paper, one workaround to obtain fine-grained gradients (subgradients over one example, or a group of examples of size smaller than batch size) is to leverage the gradient accumulation step, especially in frameworks that do not support per-example gradient computation like PyTorch.

```
for epoch in range(num_epochs):
    epoch_step = 0
    for batch in grab_ordered_batches:
        epoch_step += 1
        optimizer.zero_grad()
        grad = backward(batch)
        ... # GraB related steps
        grad_buffer.add_(grad / grad_accumulation_step)
        if epoch_step % grad_accumulation_step == 0:
            optimizer.apply_gradient(grad_buffer)
            grad_buffer.zero_()
```
Listing 1: A simple workaround to obtain fine-grained gradients in some ML frameworks such as PyTorch.

**The effect of different balancing algorithms.** We extend the experiment from Figure 1 and run Algorithm 5 and Algorithm 6 for different epochs. Figure 4 summarizes the results: although the

---

[6]https://github.com/pytorch/examples/tree/main/word_language_model
[7]https://github.com/google-research/bert.

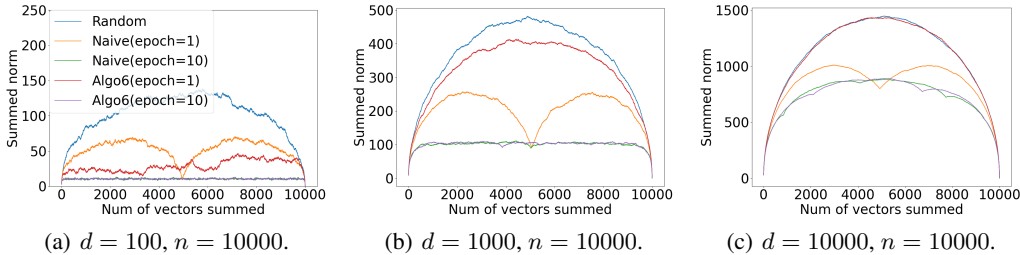

(a) $d = 100$, $n = 10000$.  (b) $d = 1000$, $n = 10000$.  (c) $d = 10000$, $n = 10000$.

Figure 4: Evaluating the herding bound of Algorithm 5 and Algorithm 6. The epoch denotes the number of times that recursively call the algorithm.

two algorithms differ when single time sorting is used, but they obtain similar results when called repeatedly (10 times). Note that the running of Algorithm 6 requires tuning a hyperparameter $c$ and generate a series of random numbers. To avoid these overhead, in practice we recommend using Algorithm 5. Another interesting observation is that in terms of $\ell$-2 norm, the naive balancing (Algorithm 5) outperforms Algorithm 6 in high-dimensional setting (b)(c) in Figure 4 when epoch= 1.

## B   Theoretical Analysis

In this section, we provide the detailed proof for all the theorems in the main paper. Throughout the proof, for simplicity, we denote $\boldsymbol{w}_{k+1} = \boldsymbol{w}_{k+1}^{(1)} = \boldsymbol{w}_k^{(n+1)}$ for all the $k \geq 1$. Additionally, we define the maximum backward deviation within an epoch to be

$$\Delta_k = \max_{m=2,\ldots,n+1} \left\| \boldsymbol{w}_k^{(m)} - \boldsymbol{w}_k \right\|_\infty \quad \text{for all} \quad k = 1, \ldots, K.$$

For any $k$, we let $\sigma_k^{-1}(i) = \{t = 1, \cdots, n \,|\, \sigma_k(t) = i\}$ to be the step when $i$-th example is visited. Unless otherwise specified, we use $\|\cdot\|$ to denote the $\ell_2$-norm.

### B.1   Details to the greedy statement

*Proof.* [25] We first construct a group of 2-d vectors. Without the loss of generality, we let $n$ divides 2. And let $n/2$ vectors be $[1, 1]^\top$ and the other $n/2$ vectors be $[4, -2]^\top$. Algorithm 1 will always select $[1, 1]^\top$ in the first $n/2$ steps based on the current sum being $[m, m]^\top, \forall m \leq n/2$. We show this with induction. Note that when $t = 1$, $[1, 1]^\top$ is selected. Then suppose in the $k$-th selection the current sum is $[k, k]^\top$, the algorithm will again select $[1, 1]^\top$ since $2(k + 1)^2 < (k + 4)^2 + (k - 2)^2$. This makes the herding objective $\Omega(n)$ with Algorithm 1.

On the other hand, consider using a random permutation. Let random variable $X_t$ denote the value of herding objective at $t$-th selection. Then we know that $X_t$ is a 2-d martingale. So that from Azuma-Hoeffding Inequality we know $\|X_k\| \leq O(\sqrt{n})$ for all $k$. $\qquad\square$

### B.2   Proof to Theorem 1

**Theorem 1.** *In Algorithm 2, under Assumptions 1, 3 and 4, if we set $\alpha$ to be*

$$\alpha = \min \left\{ \sqrt[3]{\frac{f(\boldsymbol{w}_1) - f^*}{24nH^2\varsigma^2 L_{2,\infty}^2 K}}, \frac{1}{8n(L + L_{2,\infty})}, \frac{1}{32nL_\infty}, \frac{1}{16HL_{2,\infty}} \right\}$$

*then it converges at the rate*

$$\frac{1}{K}\sum_{k=1}^{K} \|\nabla f(\boldsymbol{w}_k)\|^2 \leq 36\sqrt[3]{\frac{H^2\varsigma^2 L_{2,\infty}^2 F^2}{n^2 K^2}} + \frac{\varsigma^2}{K} + \frac{32F(L + L_{2,\infty} + L_\infty)}{K} + \frac{64FHL_{2,\infty}}{nK},$$

*where $F = f(\boldsymbol{w}_1) - f^*$. Furthermore, under the additional PL condition (Assumption 2), with $\alpha$ as*

$$\alpha = \min \left\{ \frac{1}{n\mu}, \frac{1}{48HL_{2,\infty}}, \frac{1}{96n(L + L_{2,\infty} + L_\infty)}, \frac{2}{n\mu K}W_0\left(\frac{(f(\boldsymbol{w}_1) - f^* + \varsigma^2)n^2\mu^3 K^2}{192H^2 L_{2,\infty}^2\varsigma^2}\right) \right\},$$

*where $W_0$ denotes the Lambert W function, Algorithm 2 converges at the rate*

$$f(\boldsymbol{w}_K) - f^* \leq \tilde{O}\left(\frac{H^2 L_{2,\infty}^2 \varsigma^2}{\mu^3 n^2 K^2}\right).$$

*Proof.* From Lemma 1, we get

$$\frac{1}{K}\sum_{k=1}^{K}\|\nabla f(\boldsymbol{w}_k)\|^2 \leq \frac{2(f(\boldsymbol{w}_1) - f^*)}{\alpha n K} + \frac{L_{2,\infty}^2}{K}\sum_{k=1}^{K}\max_t \left\|\boldsymbol{w}_k - \boldsymbol{w}_k^{(t)}\right\|_\infty^2.$$

On the other hand, from Lemma 3, we obtain

$$\sum_{k=1}^{K}\Delta_k^2 \leq 16\alpha^2 n^2 \varsigma^2 + 48\alpha^2 H^2 \varsigma^2 K + 48\alpha^2 n^2 \sum_{k=1}^{K}\|\nabla f(\boldsymbol{w}_k)\|_\infty^2.$$

Combining them together gives us,

$$\frac{1}{K}\sum_{k=1}^{K}\|\nabla f(\boldsymbol{w}_k)\|^2 \leq \frac{2(f(\boldsymbol{w}_1) - f^*)}{\alpha n K} + \frac{L_{2,\infty}^2}{K}\left(16\alpha^2 n^2 \varsigma^2 + 48\alpha^2 H^2 \varsigma^2 K + 48\alpha^2 n^2 \sum_{k=1}^{K}\|\nabla f(\boldsymbol{w}_k)\|_\infty^2\right)$$

$$\leq \frac{2(f(\boldsymbol{w}_1) - f^*)}{\alpha n K} + \frac{16\alpha^2 n^2 \varsigma^2 L_{2,\infty}^2}{K} + 48\alpha^2 H^2 \varsigma^2 L_{2,\infty}^2 + \frac{48\alpha^2 n^2 L_{2,\infty}^2}{K}\sum_{k=1}^{K}\|\nabla f(\boldsymbol{w}_k)\|_\infty^2.$$

Note that for any $\boldsymbol{x} \in \mathbb{R}^d$, $\|\boldsymbol{x}\|_\infty \leq \|\boldsymbol{x}\|_2$. And so the last term can by bounded by its $\ell_2$-norm. Moving it to the left side of the inequality gives us

$$\frac{1 - 48\alpha^2 n^2 L_{2,\infty}^2}{K}\sum_{k=1}^{K}\|\nabla f(\boldsymbol{w}_k)\|^2 \leq \frac{2(f(\boldsymbol{w}_1) - f^*)}{\alpha n K} + \frac{16\alpha^2 n^2 \varsigma^2 L_{2,\infty}^2}{K} + 48\alpha^2 H^2 \varsigma^2 L_{2,\infty}^2.$$

Finally, we choose $\alpha$ so that

$$\alpha = \min\left\{\sqrt[3]{\frac{f(\boldsymbol{w}_1) - f^*}{24 n H^2 \varsigma^2 L_{2,\infty}^2 K}}, \frac{1}{8n(L + L_{2,\infty})}, \frac{1}{32 n L_\infty}, \frac{1}{16 H L_{2,\infty}}\right\}$$

$$\frac{1}{K}\sum_{k=1}^{K}\|\nabla f(\boldsymbol{w}_k)\|^2 \leq \frac{4(f(\boldsymbol{w}_1) - f^*)}{\alpha n K} + \frac{32\alpha^2 n^2 \varsigma^2 L_{2,\infty}^2}{K} + 96\alpha^2 H^2 \varsigma^2 L_{2,\infty}^2$$

$$\leq 36\sqrt[3]{\frac{H^2 \varsigma^2 L_{2,\infty}^2 (f(\boldsymbol{w}_1) - f^*)^2}{n^2 K^2}} + \frac{\varsigma^2}{K}$$

$$+ \frac{32(f(\boldsymbol{w}_1) - f^*)(L + L_{2,\infty} + L_\infty)}{K} + \frac{64(f(\boldsymbol{w}_1) - f^*) H L_{2,\infty}}{n K}.$$

Denoting $F = f(\boldsymbol{w}_1) - f^*$, we finally obtain This gives

$$\frac{1}{K}\sum_{k=1}^{K}\|\nabla f(\boldsymbol{w}_k)\|^2 \leq 36\sqrt[3]{\frac{H^2 \varsigma^2 L_{2,\infty}^2 F^2}{n^2 K^2}} + \frac{\varsigma^2}{K} + \frac{32F(L + L_{2,\infty} + L_\infty)}{K} + \frac{64 F H L_{2,\infty}}{n K}.$$

This gives us the bound for general smooth-convex case. We proceed to prove the PL case:

From Lemma 3 we have the following relation:

$$\Delta_k \leq 2\alpha H\varsigma + (8\alpha n L_\infty + 4\alpha H L_{2,\infty})\Delta_{k-1} + 2\alpha n\|\nabla f(\boldsymbol{w}_k)\|_\infty, \forall k \geq 2.$$

Square on both sides

$$\Delta_k^2 \leq 3\alpha^2(4H L_{2,\infty} + 8n L_\infty)^2 \Delta_{k-1}^2 + 12\alpha^2 H^2 \varsigma^2 + 12\alpha^2 n^2\|\nabla f(\boldsymbol{w}_k)\|^2.$$

Summing from $k = 1$ to $K - 1$,

$$\sum_{k=1}^{K-1} \rho^{K-1-k} \Delta_k^2$$

$$= \rho^{K-2} \Delta_1^2 + \sum_{k=2}^{K-1} \rho^{K-1-k} \Delta_k^2$$

$$\leq \rho^{K-2} \Delta_1^2 + \sum_{k=2}^{K-1} \rho^{K-1-k} \left( 3\alpha^2 (4HL_{2,\infty} + 8nL_\infty)^2 \Delta_{k-1}^2 + 12\alpha^2 H^2 \varsigma^2 + 12\alpha^2 n^2 \|\nabla f(\boldsymbol{w}_k)\|^2 \right)$$

$$\leq \rho^{K-2} \Delta_1^2 + 3\alpha^2 (4HL_{2,\infty} + 8nL_\infty)^2 \sum_{k=2}^{K-1} \rho^{K-1-k} \Delta_{k-1}^2 + 12\alpha^2 H^2 \varsigma^2 \sum_{i=0}^{\infty} \rho^i$$

$$+ 12\alpha^2 n^2 \sum_{k=2}^{K-1} \rho^{K-1-k} \|\nabla f(\boldsymbol{w}_k)\|^2$$

$$\leq \rho^{K-2} \Delta_1^2 + 3\alpha^2 \rho^{-1} (4HL_{2,\infty} + 8nL_\infty)^2 \sum_{k=2}^{K-1} \rho^{K-1-(k-1)} \Delta_{k-1}^2 + \frac{12\alpha^2 H^2 \varsigma^2}{1 - \rho}$$

$$+ 12\alpha^2 n^2 \sum_{k=2}^{K-1} \rho^{K-1-k} \|\nabla f(\boldsymbol{w}_k)\|^2.$$

Recall from Lemma 3 that

$$\Delta_1^2 \leq 8\alpha^2 n^2 \|\nabla f(\boldsymbol{w}_1)\|_\infty^2 + 8\alpha^2 n^2 \varsigma^2,$$

this gives us,

$$\sum_{k=1}^{K-1} \rho^{K-1-k} \Delta_k^2 \leq \rho^{K-2} (8\alpha^2 n^2 \|\nabla f(\boldsymbol{w}_1)\|_\infty^2 + 8\alpha^2 n^2 \varsigma^2) + 3\alpha^2 \rho^{-1} (4HL_{2,\infty} + 8nL_\infty)^2 \sum_{k=1}^{K-1} \rho^{K-1-k} \Delta_k^2$$

$$+ \frac{12\alpha^2 H^2 \varsigma^2}{1 - \rho} + 12\alpha^2 n^2 \sum_{k=2}^{K-1} \rho^{K-1-k} \|\nabla f(\boldsymbol{w}_k)\|^2.$$

Using the fact that

$$\alpha = \min \left\{ \frac{1}{n\mu}, \frac{1}{48HL_{2,\infty}}, \frac{1}{96n(L + L_{2,\infty} + L_\infty)} \right\}$$

Solve it, we obtain

$$\sum_{k=1}^{K-1} \rho^{K-1-k} \Delta_k^2 \leq 64 \rho^K \alpha^2 n^2 \varsigma^2 + \frac{24\alpha^2 H^2 \varsigma^2}{1 - \rho} + 24\alpha^2 n^2 \sum_{k=1}^{K-1} \rho^{K-1-k} \|\nabla f(\boldsymbol{w}_k)\|^2.$$

First from Lemma 2 we get

$$f(\boldsymbol{w}_K) - f^* \leq \rho^K (f(\boldsymbol{w}_1) - f^*) + \frac{\alpha n}{2} L_{2,\infty}^2 \sum_{k=1}^{K-1} \rho^{K-1-k} \Delta_k^2 - \frac{\alpha n}{4} \sum_{k=1}^{K-1} \rho^{K-1-k} \|\nabla f(\boldsymbol{w}_k)\|^2$$

$$\leq \rho^K (f(\boldsymbol{w}_1) - f^*) + 32 \rho^K \alpha^3 n^3 L_{2,\infty}^2 \varsigma^2 + \frac{24\alpha^2 H^2 L_{2,\infty}^2 \varsigma^2}{\mu},$$

where in the last step, we apply the learning rate bound that

$$\alpha \leq \frac{1}{96n(L + L_{2,\infty} + L_\infty)} \leq \frac{1}{96nL_{2,\infty}}.$$

The RHS of the inequality can be further bounded by

$$(1 - \alpha n\mu/2)^K (f(\boldsymbol{w}_1) - f^* + \varsigma^2) + \frac{24\alpha^2 H^2 L_{2,\infty}^2 \varsigma^2}{\mu}$$

$$\leq (f(\boldsymbol{w}_1) - f^* + \varsigma^2) \exp(-\alpha n \mu K / 2) + 24\alpha^2 H^2 L_{2,\infty}^2 \varsigma^2 \mu^{-1}.$$

Take derivative with respect to $\alpha$ and set it to zero, we obtain

$$\alpha = \frac{2}{n\mu K} W_0 \left( \frac{(f(\boldsymbol{w}_1) - f^* + \varsigma^2) n^2 \mu^3 K^2}{192 H^2 L_{2,\infty}^2 \varsigma^2} \right),$$

and

$$f(\boldsymbol{w}_K) - f^* \leq \frac{288 H^2 L_{2,\infty}^2 \varsigma^2}{n^2 \mu^3 K^2} \cdot W_0 \left( \frac{(f(\boldsymbol{w}_1) - f^* + \varsigma^2) n^2 \mu^3 K^2}{192 H^2 L_{2,\infty}^2 \varsigma^2} \right) \left[ 1 + W_0 \left( \frac{(f(\boldsymbol{w}_1) - f^* + \varsigma^2) n^2 \mu^3 K^2}{192 H^2 L_{2,\infty}^2 \varsigma^2} \right) \right]$$

$$= \tilde{O} \left( \frac{H^2 L_{2,\infty}^2 \varsigma^2}{\mu^3 n^2 K^2} \right),$$

where $W_0(\cdot)$ denotes the Lambert W function. That completes the proof. $\qquad\square$

### B.3 Proof to Theorem 3

**Theorem 3.** *In Algorithm 4, under Assumptions 1, 3 and 5, if we set $\alpha$ to be*

$$\alpha = \min \left\{ \sqrt[3]{\frac{f(\boldsymbol{w}_1) - f^*}{32 n A^2 \varsigma^2 L_{2,\infty}^2 K}}, \frac{1}{nL}, \frac{1}{26(n+A)L_{2,\infty}}, \frac{1}{260 n L_\infty} \right\}$$

*then it converges at the rate*

$$\frac{1}{K} \sum_{k=1}^{K} \|\nabla f(\boldsymbol{w}_k)\|^2 \leq 11 \sqrt[3]{\frac{H^2 \varsigma^2 L_{2,\infty}^2 F^2}{n^2 K^2}} + \frac{\varsigma^2}{K} + \frac{65(F(L + L_{2,\infty} + L_\infty))}{K} + \frac{8 F A L_{2,\infty}}{nK}.$$

*where $F = f(\boldsymbol{w}_1) - f^*$. Furthermore, under the additional PL condition (Assumption 2) with $\alpha$ as*

$$\alpha = \min \left\{ \frac{1}{n\mu}, \frac{1}{nL}, \frac{1}{52(n+A)L_{2,\infty}}, \frac{1}{520 n L_\infty}, \frac{2}{n\mu K} W_0 \left( \frac{(f(\boldsymbol{w}_1) - f^* + \varsigma^2) n^2 \mu^3 K^2}{256 A^2 L_{2,\infty}^2 \varsigma^2} \right) \right\},$$

*where $W_0$ denotes the Lambert W function, it converges at the rate*

$$f(\boldsymbol{w}_K) - f^* \leq \tilde{O} \left( \frac{A^2 L_{2,\infty}^2 \varsigma^2}{\mu^3 n^2 K^2} \right).$$

*Proof.* From Lemma 1, we get

$$\frac{1}{K} \sum_{k=1}^{K} \|\nabla f(\boldsymbol{w}_k)\|^2 \leq \frac{2(f(\boldsymbol{w}_1) - f^*)}{\alpha n K} + \frac{L_{2,\infty}^2}{K} \sum_{k=1}^{K} \max_t \left\| \boldsymbol{w}_k - \boldsymbol{w}_k^{(t)} \right\|_\infty^2.$$

On the other hand, from Lemma 4, we obtain

$$\sum_{k=1}^{K} \Delta_k^2 \leq 120 \alpha^2 n^2 \varsigma^2 + 64 \alpha^2 A^2 \varsigma^2 K + 48 \alpha^2 n^2 \sum_{k=1}^{K} \|\nabla f(\boldsymbol{w}_k)\|_\infty^2.$$

Combining them together gives us,

$$\frac{1}{K} \sum_{k=1}^{K} \|\nabla f(\boldsymbol{w}_k)\|^2$$

$$\leq \frac{2(f(\boldsymbol{w}_1) - f^*)}{\alpha n K} + \frac{L_{2,\infty}^2}{K} \left( 120 \alpha^2 n^2 \varsigma^2 + 64 \alpha^2 A^2 \varsigma^2 K + 48 \alpha^2 n^2 \sum_{k=1}^{K} \|\nabla f(\boldsymbol{w}_k)\|_\infty^2 \right)$$

$$\leq \frac{2(f(\boldsymbol{w}_1) - f^*)}{\alpha n K} + \frac{120 \alpha^2 n^2 \varsigma^2 L_{2,\infty}^2}{K} + 64 \alpha^2 A^2 \varsigma^2 L_{2,\infty}^2 + \frac{48 \alpha^2 n^2 L_{2,\infty}^2}{K} \sum_{k=1}^{K} \|\nabla f(\boldsymbol{w}_k)\|^2.$$

Given

$$\alpha = \min\left\{\sqrt[3]{\frac{f(\boldsymbol{w}_1) - f^*}{32nA^2\varsigma^2 L_{2,\infty}^2 K}}, \frac{1}{nL}, \frac{1}{26(n+A)L_{2,\infty}}, \frac{1}{260nL_\infty}\right\},$$

we get

$$\frac{1}{K}\sum_{k=1}^{K}\|\nabla f(\boldsymbol{w}_k)\|^2 \leq \frac{4(f(\boldsymbol{w}_1) - f^*)}{\alpha nK} + \frac{240\alpha^2 n^2\varsigma^2 L_{2,\infty}^2}{K} + 128\alpha^2 A^2\varsigma^2 L_{2,\infty}^2$$

$$\leq 11\sqrt[3]{\frac{H^2\varsigma^2 L_{2,\infty}^2(f(\boldsymbol{w}_1) - f^*)^2}{n^2 K^2}} + \frac{\varsigma^2}{K}$$

$$+ \frac{65(f(\boldsymbol{w}_1) - f^*)(L + L_{2,\infty} + L_\infty)}{K} + \frac{8(f(\boldsymbol{w}_1) - f^*)AL_{2,\infty}}{nK}.$$

This gives us the bound for general smooth-convex case. We proceed to prove the PL case: From Lemma 7 we know

$$\sum_{k=1}^{K-1}\rho^{K-1-k}\Delta_k^2 \leq 1024\alpha^2\rho^K n^2\varsigma^2 + \frac{120\alpha^2 A^2\varsigma^2}{1-\rho} + 64\alpha^2 n^2 \sum_{k=1}^{K-1}\rho^{K-1-k}\|\nabla f(\boldsymbol{w}_k)\|_\infty^2.$$

On the other hand, from Lemma 2 we get

$$f(\boldsymbol{w}_K) - f^* \leq \rho^K(f(\boldsymbol{w}_1) - f^*) + \frac{\alpha n}{2}L_{2,\infty}^2\sum_{k=1}^{K-1}\rho^{K-1-k}\Delta_k^2 - \frac{\alpha n}{4}\sum_{k=1}^{K-1}\rho^{K-1-k}\|\nabla f(\boldsymbol{w}_k)\|^2$$

$$\leq \rho^K(f(\boldsymbol{w}_1) - f^*) + 512\rho^K\alpha^3 n^3 L_{2,\infty}^2\varsigma^2 + \frac{120\alpha^2 A^2 L_{2,\infty}^2\varsigma^2}{\mu},$$

where in the last step, we apply the learning rate bound that

$$\alpha \leq \frac{1}{52nL_{2,\infty}}.$$

The RHS of the inequality can be further bounded by

$$(1 - \alpha n\mu/2)^K(f(\boldsymbol{w}_1) - f^* + \varsigma^2) + \frac{120\alpha^2 A^2 L_{2,\infty}^2\varsigma^2}{\mu}$$

$$\leq (f(\boldsymbol{w}_1) - f^* + \varsigma^2)\exp(-\alpha n\mu K/2) + 120\alpha^2 A^2 L_{2,\infty}^2\varsigma^2\mu^{-1}.$$

Take derivative with respect to $\alpha$ and set it to zero, we obtain

$$\alpha = \frac{2}{n\mu K}W_0\left(\frac{(f(\boldsymbol{w}_1) - f^* + \varsigma^2)n^2\mu^3 K^2}{256A^2 L_{2,\infty}^2\varsigma^2}\right),$$

and

$$f(\boldsymbol{w}_K) - f^* \leq \frac{320A^2 L_{2,\infty}^2\varsigma^2}{n^2\mu^3 K^2} \cdot W_0\left(\frac{(f(\boldsymbol{w}_1) - f^* + \varsigma^2)n^2\mu^3 K^2}{256A^2 L_{2,\infty}^2\varsigma^2}\right)\left[1 + W_0\left(\frac{(f(\boldsymbol{w}_1) - f^* + \varsigma^2)n^2\mu^3 K^2}{256A^2 L_{2,\infty}^2\varsigma^2}\right)\right]$$

$$= \tilde{O}\left(\frac{A^2 L_{2,\infty}^2\varsigma^2}{\mu^3 n^2 K^2}\right),$$

where $W_0(\cdot)$ denotes the Lambert W function. That completes the proof. $\square$

## B.4 Technical Lemmas

**Lemma 1.** *In Algorithm 2 and Algorithm 4, if $\alpha nL < 1$ holds and Assumption 1 (except PL condition), 3 and 4 hold, then*

$$\frac{1}{K}\sum_{k=1}^{K}\|\nabla f(\boldsymbol{w}_k)\|^2 \leq \frac{2(f(\boldsymbol{w}_1) - f^*)}{\alpha nK} + \frac{L_{2,\infty}^2}{K}\sum_{k=1}^{K}\max_t\left\|\boldsymbol{w}_k - \boldsymbol{w}_k^{(t)}\right\|_\infty^2.$$

*Proof.* Note that in both algorithms, the update can be written as

$$\boldsymbol{w}_{k+1} = \boldsymbol{w}_k - \alpha \sum_{t=1}^{n} \nabla f(\boldsymbol{w}_k^{(t)}; \boldsymbol{x}_{\sigma_k(t)}).$$

By the Taylor Theorem, for all the $k = 1, \cdots, K-1$,

$$f(\boldsymbol{w}_{k+1})$$

$$\leq f(\boldsymbol{w}_k) - \alpha n \left\langle \nabla f(\boldsymbol{w}_k), \frac{1}{n} \sum_{t=1}^{n} \nabla f(\boldsymbol{w}_k^{(t)}; \boldsymbol{x}_{\sigma_k(t)}) \right\rangle + \frac{\alpha^2 n^2 L}{2} \left\| \frac{1}{n} \sum_{t=1}^{n} \nabla f(\boldsymbol{w}_k^{(t)}; \boldsymbol{x}_{\sigma_k(t)}) \right\|^2$$

$$= f(\boldsymbol{w}_k) - \frac{\alpha n}{2} \|\nabla f(\boldsymbol{w}_k)\|^2 - \frac{\alpha n}{2} \left\| \frac{1}{n} \sum_{t=1}^{n} \nabla f(\boldsymbol{w}_k^{(t)}; \boldsymbol{x}_{\sigma_k(t)}) \right\|^2$$

$$+ \frac{\alpha n}{2} \left\| \nabla f(\boldsymbol{w}_k) - \frac{1}{n} \sum_{t=1}^{n} \nabla f(\boldsymbol{w}_k^{(t)}; \boldsymbol{x}_{\sigma_k(t)}) \right\|^2 + \frac{\alpha^2 n^2 L}{2} \left\| \frac{1}{n} \sum_{t=1}^{n} \nabla f(\boldsymbol{w}_k^{(t)}; \boldsymbol{x}_{\sigma_k(t)}) \right\|^2$$

$$\leq f(\boldsymbol{w}_k) - \frac{\alpha n}{2} \|\nabla f(\boldsymbol{w}_k)\|^2 + \frac{\alpha n}{2} \left\| \nabla f(\boldsymbol{w}_k) - \frac{1}{n} \sum_{t=1}^{n} \nabla f(\boldsymbol{w}_k^{(t)}; \boldsymbol{x}_{\sigma_k(t)}) \right\|^2.$$

In the second step, we apply $-\langle \boldsymbol{a}, \boldsymbol{b} \rangle = -\frac{1}{2}\|\boldsymbol{a}\|^2 - \frac{1}{2}\|\boldsymbol{b}\|^2 + \frac{1}{2}\|\boldsymbol{a} - \boldsymbol{b}\|^2, \forall \boldsymbol{a}, \boldsymbol{b}$. In the third step, we use the condition that $\alpha n L < 1$. Expanding the last term using Assumption 1, we get

$$\left\| \nabla f(\boldsymbol{w}_k) - \frac{1}{n} \sum_{t=1}^{n} \nabla f(\boldsymbol{w}_k^{(t)}; \boldsymbol{x}_{\sigma_k(t)}) \right\|^2 = \left\| \frac{1}{n} \sum_{t=1}^{n} \nabla f(\boldsymbol{w}_k; \boldsymbol{x}_{\sigma_k(t)})) - \frac{1}{n} \sum_{t=1}^{n} \nabla f(\boldsymbol{w}_k^{(t)}; \boldsymbol{x}_{\sigma_k(t)}) \right\|^2$$

$$\leq \frac{1}{n} \sum_{t=1}^{n} \left\| \nabla f(\boldsymbol{w}_k; \boldsymbol{x}_{\sigma_k(t)})) - \nabla f(\boldsymbol{w}_k^{(t)}; \boldsymbol{x}_{\sigma_k(t)}) \right\|^2$$

$$\leq \frac{L_{2,\infty}^2}{n} \sum_{t=1}^{n} \left\| \boldsymbol{w}_k - \boldsymbol{w}_k^{(t)} \right\|_{\infty}^2$$

$$\leq L_{2,\infty}^2 \Delta_k^2.$$

In the second step we apply the Jensen Inequality. Put it back, we obtain

$$f(\boldsymbol{w}_{k+1}) \leq f(\boldsymbol{w}_k) - \frac{\alpha n}{2} \|\nabla f(\boldsymbol{w}_k)\|^2 + \frac{\alpha n L_{2,\infty}^2 \Delta_k^2}{2}.$$

Finally, summing from $k = 1$ to $K-1$, we obtain

$$\frac{1}{K} \sum_{k=1}^{K} \|\nabla f(\boldsymbol{w}_k)\|^2 \leq \frac{2(f(\boldsymbol{w}_1) - f^*)}{\alpha n K} + \frac{L_{2,\infty}^2}{K} \sum_{k=1}^{K} \max_t \left\| \boldsymbol{w}_k - \boldsymbol{w}_k^{(t)} \right\|_{\infty}^2.$$

That completes the proof. $\qquad\square$

**Lemma 2.** *In Algorithm 2 and Algorithm 4, if $\alpha n L < 1$ holds and Assumption 1 (including PL condition), 3 and 4 hold, then*

$$f(\boldsymbol{w}_K) - f^* \leq \rho^K (f(\boldsymbol{w}_1) - f^*) + \frac{\alpha n}{2} L_{2,\infty}^2 \sum_{k=1}^{K-1} \rho^{K-1-k} \Delta_k^2 - \frac{\alpha n}{4} \sum_{k=1}^{K-1} \rho^{K-1-k} \|\nabla f(\boldsymbol{w}_k)\|^2.$$

*Proof.* Since the this lemma is a special case of Lemma 1, we can just borrow the derivation there and get for all the $k = 1, \cdots, K-1$

$$f(\boldsymbol{w}_{k+1}) \leq f(\boldsymbol{w}_k) + \frac{\alpha n}{2} L_{2,\infty}^2 \Delta_k^2 - \frac{\alpha n}{2} \|\nabla f(\boldsymbol{w}_k)\|^2$$

$$= f(\boldsymbol{w}_k) + \frac{\alpha n}{2} L_{2,\infty}^2 \Delta_k^2 - \frac{\alpha n}{4} \|\nabla f(\boldsymbol{w}_k)\|^2 - \frac{\alpha n}{4} \|\nabla f(\boldsymbol{w}_k)\|^2$$

$$\leq f(\boldsymbol{w}_k) + \frac{\alpha n}{2} L_{2,\infty}^2 \Delta_k^2 - \frac{\alpha n \mu}{2}(f(\boldsymbol{w}_k) - f^*) - \frac{\alpha n}{4} \|\nabla f(\boldsymbol{w}_k)\|^2. \quad \text{(PL condition)}$$

Define $1 - \alpha n \mu/2 = \rho$, then

$$f(\boldsymbol{w}_{k+1}) - f^* \leq \rho(f(\boldsymbol{w}_k) - f^*) + \frac{\alpha n}{2} L_{2,\infty}^2 \Delta_k^2 - \frac{\alpha n}{4} \|\nabla f(\boldsymbol{w}_k)\|^2.$$

Recursively apply it from $k = 1$ to $K - 1$, we obtain

$$f(\boldsymbol{w}_K) - f^* \leq \rho^K (f(\boldsymbol{w}_1) - f^*) + \frac{\alpha n}{2} L_{2,\infty}^2 \sum_{k=1}^{K-1} \rho^{K-1-k} \Delta_k^2 - \frac{\alpha n}{4} \sum_{k=1}^{K-1} \rho^{K-1-k} \|\nabla f(\boldsymbol{w}_k)\|^2.$$

That completes the proof. $\qquad\qquad\qquad\qquad\qquad\qquad\qquad\qquad\qquad\qquad\qquad\qquad\qquad\qquad\square$

**Lemma 3.** *In Algorithm 2, if the learning rate $\alpha$ fulfills*

$$\alpha \leq \min\left\{\frac{1}{32nL_\infty}, \frac{1}{16HL_{2,\infty}}\right\},$$

*then the following inequalities hold:*

$$\Delta_k \leq 2\alpha H\varsigma + (8\alpha n L_\infty + 4\alpha H L_{2,\infty})\Delta_{k-1} + 2\alpha n \|\nabla f(\boldsymbol{w}_k)\|_\infty, \forall k \geq 2$$

*and,*

$$\Delta_1^2 \leq 8\alpha^2 n^2 \|\nabla f(\boldsymbol{w}_1)\|_\infty^2 + 8\alpha^2 n^2 \varsigma^2,$$

*and finally,*

$$\sum_{k=1}^{K} \Delta_k^2 \leq 16\alpha^2 n^2 \varsigma^2 + 48\alpha^2 H^2 \varsigma^2 K + 48\alpha^2 n^2 \sum_{k=1}^{K} \|\nabla f(\boldsymbol{w}_k)\|_\infty^2.$$

*Proof.* Without the loss of generality, for all the $m \in \{2, \cdots, n+1\}$ and all the $k \in \{2, \cdots, K\}$,

$$\boldsymbol{w}_k^{(m)} = \boldsymbol{w}_k - \alpha \sum_{t=1}^{m-1} \nabla f\left(\boldsymbol{w}_k^{(t)}; \boldsymbol{x}_{\sigma_k(t)}\right)$$

$$= \boldsymbol{w}_k - \alpha \sum_{t=1}^{m-1} \nabla f\left(\boldsymbol{w}_{k-1}^{(\sigma_{k-1}^{-1}(\sigma_k(t)))}; \boldsymbol{x}_{\sigma_k(t)}\right)$$

$$- \alpha \sum_{t=1}^{m-1} \left(\nabla f\left(\boldsymbol{w}_k^{(t)}; \boldsymbol{x}_{\sigma_k(t)}\right) - \nabla f\left(\boldsymbol{w}_{k-1}^{(\sigma_{k-1}^{-1}(\sigma_k(t)))}; \boldsymbol{x}_{\sigma_k(t)}\right)\right).$$

Now add and subtract

$$\alpha \sum_{t=1}^{m-1} \frac{1}{n} \sum_{s=1}^{n} \nabla f\left(\boldsymbol{w}_{k-1}^{(s)}; \boldsymbol{x}_{\sigma_{k-1}(s)}\right) = \frac{\alpha(m-1)}{n} \sum_{t=1}^{n} \nabla f\left(\boldsymbol{w}_{k-1}^{(t)}; \boldsymbol{x}_{\sigma_{k-1}(t)}\right),$$

which gives

$$\boldsymbol{w}_k^{(m)} = \boldsymbol{w}_k - \alpha \sum_{t=1}^{m-1} \left(\nabla f\left(\boldsymbol{w}_{k-1}^{(\sigma_{k-1}^{-1}(\sigma_k(t)))}; \boldsymbol{x}_{\sigma_k(t)}\right) - \frac{1}{n} \sum_{s=1}^{n} \nabla f\left(\boldsymbol{w}_{k-1}^{(s)}; \boldsymbol{x}_{\sigma_{k-1}(s)}\right)\right)$$

$$- \frac{\alpha(m-1)}{n} \sum_{t=1}^{n} \nabla f\left(\boldsymbol{w}_{k-1}^{(t)}; \boldsymbol{x}_{\sigma_{k-1}(t)}\right)$$

$$- \alpha \sum_{t=1}^{m-1} \left(\nabla f\left(\boldsymbol{w}_k^{(t)}; \boldsymbol{x}_{\sigma_k(t)}\right) - \nabla f\left(\boldsymbol{w}_{k-1}^{(\sigma_{k-1}^{-1}(\sigma_k(t)))}; \boldsymbol{x}_{\sigma_k(t)}\right)\right).$$

We further add and subtract

$$\frac{\alpha(m-1)}{n} \sum_{t=1}^{n} \nabla f(\boldsymbol{w}_k; \boldsymbol{x}_{\sigma_{k-1}(t)}) = \alpha(m-1)\nabla f(\boldsymbol{w}_k)$$

to arrive at

$$\boldsymbol{w}_k^{(m)} = \boldsymbol{w}_k - \alpha \sum_{t=1}^{m-1} \left( \nabla f \left( \boldsymbol{w}_{k-1}^{(\sigma_{k-1}^{-1}(\sigma_k(t)))}; \boldsymbol{x}_{\sigma_k(t)} \right) - \frac{1}{n} \sum_{s=1}^{n} \nabla f \left( \boldsymbol{w}_{k-1}^{(s)}; \boldsymbol{x}_{\sigma_{k-1}(s)} \right) \right)$$

$$- \alpha(m-1)\nabla f(\boldsymbol{w}_k) + \frac{\alpha(m-1)}{n} \sum_{t=1}^{n} \left( \nabla f \left( \boldsymbol{w}_k; \boldsymbol{x}_{\sigma_{k-1}(t)} \right) - \nabla f \left( \boldsymbol{w}_{k-1}^{(t)}; \boldsymbol{x}_{\sigma_{k-1}(t)} \right) \right)$$

$$- \alpha \sum_{t=1}^{m-1} \left( \nabla f \left( \boldsymbol{w}_k^{(t)}; \boldsymbol{x}_{\sigma_k(t)} \right) - \nabla f \left( \boldsymbol{w}_{k-1}^{(\sigma_{k-1}^{-1}(\sigma_k(t)))}; \boldsymbol{x}_{\sigma_k(t)} \right) \right).$$

We can now re-arrange, take norms on both sides and apply the triangle inequality,

$$\left\| \boldsymbol{w}_k^{(m)} - \boldsymbol{w}_k \right\|_\infty \leq \alpha \left\| \sum_{t=1}^{m-1} \left( \nabla f \left( \boldsymbol{w}_{k-1}^{(\sigma_{k-1}^{-1}(\sigma_k(t)))}; \boldsymbol{x}_{\sigma_k(t)} \right) - \frac{1}{n} \sum_{s=1}^{n} \nabla f \left( \boldsymbol{w}_{k-1}^{(s)}; \boldsymbol{x}_{\sigma_{k-1}(s)} \right) \right) \right\|_\infty$$

$$+ \alpha(m-1)\|\nabla f(\boldsymbol{w}_k)\|_\infty$$

$$+ \frac{\alpha(m-1)}{n} \left\| \sum_{t=1}^{n} \left( \nabla f \left( \boldsymbol{w}_k; \boldsymbol{x}_{\sigma_{k-1}(t)} \right) - \nabla f \left( \boldsymbol{w}_{k-1}^{(t)}; \boldsymbol{x}_{\sigma_{k-1}(t)} \right) \right) \right\|_\infty$$

$$+ \alpha \left\| \sum_{t=1}^{m-1} \left( \nabla f \left( \boldsymbol{w}_k^{(t)}; \boldsymbol{x}_{\sigma_k(t)} \right) - \nabla f \left( \boldsymbol{w}_{k-1}^{(\sigma_{k-1}^{-1}(\sigma_k(t)))}; \boldsymbol{x}_{\sigma_k(t)} \right) \right) \right\|_\infty. \quad (4)$$

There are four different terms on the right hand side, we will apply the Assumption 4 on the first term, and Assumption 1 on the last two terms. First, for the first term,

$$\left\| \nabla f \left( \boldsymbol{w}_{k-1}^{(\sigma_{k-1}^{-1}(\sigma_k(t)))}; \boldsymbol{x}_{\sigma_k(t)} \right) - \frac{1}{n} \sum_{s=1}^{n} \nabla f \left( \boldsymbol{w}_{k-1}^{(s)}; \boldsymbol{x}_{\sigma_{k-1}(s)} \right) \right\|$$

$$\leq \left\| \nabla f \left( \boldsymbol{w}_{k-1}^{(\sigma_{k-1}^{-1}(\sigma_k(t)))}; \boldsymbol{x}_{\sigma_k(t)} \right) - \frac{1}{n} \sum_{s=1}^{n} \nabla f \left( \boldsymbol{w}_{k-1}^{(\sigma_{k-1}^{-1}(\sigma_k(t)))}; \boldsymbol{x}_{\sigma_{k-1}(s)} \right) \right\|$$

$$+ \left\| \frac{1}{n} \sum_{s=1}^{n} \nabla f \left( \boldsymbol{w}_{k-1}^{(\sigma_{k-1}^{-1}(\sigma_k(t)))}; \boldsymbol{x}_{\sigma_{k-1}(s)} \right) - \frac{1}{n} \sum_{s=1}^{n} \nabla f \left( \boldsymbol{w}_{k-1}^{(s)}; \boldsymbol{x}_{\sigma_{k-1}(s)} \right) \right\|$$

$$\overset{\text{Assume. 1 and 3}}{\leq} \varsigma + \frac{L_{2,\infty}}{n} \sum_{s=1}^{n} \left\| \boldsymbol{w}_{k-1}^{(\sigma_{k-1}^{-1}(\sigma_k(t)))} - \boldsymbol{w}_{k-1}^{(s)} \right\|_\infty$$

$$\leq \varsigma + \frac{L_{2,\infty}}{n} \sum_{s=1}^{n} \left( \left\| \boldsymbol{w}_{k-1} - \boldsymbol{w}_{k-1}^{(\sigma_{k-1}^{-1}(\sigma_k(t)))} \right\|_\infty + \left\| \boldsymbol{w}_{k-1} - \boldsymbol{w}_{k-1}^{(s)} \right\|_\infty \right)$$

$$\leq \varsigma + 2L_{2,\infty}\Delta_{k-1}$$

This implies if we denote

$$\boldsymbol{u}_t := \nabla f \left( \boldsymbol{w}_{k-1}^{\sigma_{k-1}^{-1}(\sigma_k(t))}; \boldsymbol{x}_{\sigma_k(t)} \right) - \frac{1}{n} \sum_{s=1}^{n} \nabla f(\boldsymbol{w}_{k-1}^{(s)}; \boldsymbol{x}_{\sigma_{k-1}(s)})$$

We can now use Assumption 4 to obtain a bound on the prefix sum

$$\left\| \sum_{t=1}^{m-1} \frac{\boldsymbol{u}_t}{\varsigma + 2L_{2,\infty}\Delta_{k-1}} \right\|_\infty \leq H,$$

that is,

$$\left\| \sum_{t=1}^{m-1} \left( \nabla f \left( \boldsymbol{w}_{k-1}^{(\sigma_{k-1}^{-1}(\sigma_k(t)))}; \boldsymbol{x}_{\sigma_k(t)} \right) - \frac{1}{n} \sum_{s=1}^{n} \nabla f \left( \boldsymbol{w}_{k-1}^{(s)}; \boldsymbol{x}_{\sigma_{k-1}(s)} \right) \right) \right\|_\infty \leq H(\varsigma + 2L_{2,\infty}\Delta_{k-1}).$$

Now we have a bound for the first term in Equation (4), we proceed to bound the last two terms where we apply Assumption 1. We can then rewrite Equation (4) into,

$$\left\|\boldsymbol{w}_k^{(m)} - \boldsymbol{w}_k\right\|_\infty \leq \alpha H(\varsigma + 2L_{2,\infty}\Delta_{k-1}) + \alpha(m-1)\|\nabla f(\boldsymbol{w}_k)\|_\infty + \frac{\alpha L_\infty(m-1)}{n}\sum_{t=1}^n \left\|\boldsymbol{w}_k - \boldsymbol{w}_{k-1}^{(t)}\right\|_\infty$$
$$+ \alpha L_\infty \sum_{t=1}^{m-1}\left\|\boldsymbol{w}_k^{(t)} - \boldsymbol{w}_{k-1}^{(\sigma_{k-1}^{-1}(\sigma_k(t)))}\right\|_\infty.$$

Furthermore, applying the triangle inequality to the norms in the last two terms, we obtain

$$\left\|\boldsymbol{w}_{k-1}^{(t)} - \boldsymbol{w}_k\right\|_\infty = \left\|\boldsymbol{w}_{k-1}^{(t)} - \boldsymbol{w}_{k-1} + \boldsymbol{w}_{k-1} - \boldsymbol{w}_{k-1}^{(n+1)}\right\|_\infty \leq 2\Delta_{k-1}$$

and similarly,

$$\left\|\boldsymbol{w}_k^{(t)} - \boldsymbol{w}_{k-1}^{(\sigma_{k-1}^{-1}(\sigma_k(t)))}\right\|_\infty = \left\|\boldsymbol{w}_k^{(t)} - \boldsymbol{w}_k + \boldsymbol{w}_k - \boldsymbol{w}_{k-1} + \boldsymbol{w}_{k-1} - \boldsymbol{w}_{k-1}^{(\sigma_{k-1}^{-1}(\sigma_k(t)))}\right\|_\infty \leq \Delta_k + 2\Delta_{k-1}.$$

This gives us

$$\left\|\boldsymbol{w}_k^{(m)} - \boldsymbol{w}_k\right\|_\infty \leq \alpha H(\varsigma + 2L_{2,\infty}\Delta_{k-1}) + \alpha(m-1)\|\nabla f(\boldsymbol{w}_k)\|_\infty + 2\alpha L_\infty(m-1)\Delta_{k-1}$$
$$+ \alpha L_\infty(m-1)(2\Delta_{k-1} + \Delta_k)$$
$$\leq \alpha H(\varsigma + 2L_{2,\infty}\Delta_{k-1}) + \alpha(m-1)\|\nabla f(\boldsymbol{w}_k)\|_\infty + \alpha L_\infty(m-1)(4\Delta_{k-1} + \Delta_k). \tag{5}$$

Note that Equation (5) only holds with $k \in \{2, \cdots, K\}$ and $m \in \{2, \cdots, n+1\}$. We now discuss the boundary cases. Note that the bound of Equation (5) trivially holds with $m = 1$ for any $k$ since the left hand side becomes zero. On the other hand, when $k = 1$, we have,

$$\boldsymbol{w}_1^{(m)} = \boldsymbol{w}_1 - \alpha \sum_{t=1}^{m-1}\nabla f\left(\boldsymbol{w}_1^{(t)}; \boldsymbol{x}_{\sigma_1(t)}\right)$$
$$= \boldsymbol{w}_1 - \alpha \sum_{t=1}^{m-1}\frac{1}{n}\sum_{s=1}^n \nabla f\left(\boldsymbol{w}_1; \boldsymbol{x}_{\sigma_1(s)}\right) + \alpha \sum_{t=1}^{m-1}\nabla f\left(\boldsymbol{w}_1^{(t)}; \boldsymbol{x}_{\sigma_1(t)}\right) - \alpha \sum_{t=1}^{m-1}\nabla f\left(\boldsymbol{w}_1; \boldsymbol{x}_{\sigma_1(t)}\right)$$
$$+ \alpha \sum_{t=1}^{m-1}\nabla f\left(\boldsymbol{w}_1; \boldsymbol{x}_{\sigma_1(t)}\right) - \alpha \sum_{t=1}^{m-1}\frac{1}{n}\sum_{s=1}^n \nabla f\left(\boldsymbol{w}_1; \boldsymbol{x}_{\sigma_1(s)}\right),$$

take norms and apply the triangle inequality, we obtain

$$\left\|\boldsymbol{w}_1^{(m)} - \boldsymbol{w}_1\right\|_\infty \leq \alpha\left\|\sum_{t=1}^{m-1}\frac{1}{n}\sum_{s=1}^n \nabla f\left(\boldsymbol{w}_1; \boldsymbol{x}_{\sigma_1(s)}\right)\right\|_\infty + \alpha\left\|\sum_{t=1}^{m-1}\left(\nabla f\left(\boldsymbol{w}_1^{(t)}; \boldsymbol{x}_{\sigma_1(t)}\right) - \nabla f\left(\boldsymbol{w}_1; \boldsymbol{x}_{\sigma_1(s)}\right)\right)\right\|_\infty$$
$$+ \alpha\left\|\sum_{t=1}^{m-1}\left(\nabla f\left(\boldsymbol{w}_1; \boldsymbol{x}_{\sigma_1(t)}\right) - \frac{1}{n}\sum_{s=1}^n \nabla f\left(\boldsymbol{w}_1; \boldsymbol{x}_{\sigma_1(s)}\right)\right)\right\|_\infty$$
$$\leq \alpha(m-1)\|\nabla f(\boldsymbol{w}_1)\|_\infty + \alpha(m-1)L_\infty\Delta_1 + \alpha(m-1)\varsigma$$
$$\leq \alpha n\|\nabla f(\boldsymbol{w}_1)\|_\infty + \alpha n L_\infty\Delta_1 + \alpha n\varsigma. \tag{6}$$

Now that we have the bounds for $\Delta_k$, we next will sum them up. Taking a max over $m$ on both side in Equation (5), this implies for all the $k \geq 2$,

$$\Delta_k \leq \alpha H(\varsigma + 2L_{2,\infty}\Delta_{k-1}) + \alpha L_\infty n(4\Delta_{k-1} + \Delta_k) + \alpha n\|\nabla f(\boldsymbol{w}_k)\|_\infty$$

as $m - 1 \leq n$. Considering the fact that $\alpha L_\infty n < 1/2$, we get

$$\Delta_k \leq 2\alpha H\varsigma + (8\alpha n L_\infty + 4\alpha H L_{2,\infty})\Delta_{k-1} + 2\alpha n\|\nabla f(\boldsymbol{w}_k)\|_\infty.$$

This completes the proof of the first inequality in the lemma. Applying this recursively from any $k \geq 2$ to 2, this gives

$$\Delta_k \leq (8\alpha n L_\infty + 4\alpha H L_{2,\infty})^{k-1}\Delta_1 + \sum_{i=1}^\infty (8\alpha n L_\infty + 4\alpha H L_{2,\infty})^i \left(2\alpha H\varsigma + 2\alpha n\|\nabla f(\boldsymbol{w}_k)\|_\infty\right).$$

Applying the learning rate conditions that $32\alpha n L_\infty \leq 1$ and $16\alpha H L_{2,\infty} \leq 1$, we obtain

$$\Delta_k \leq \left(\frac{1}{2}\right)^{k-1} \Delta_1 + 4\alpha H\varsigma + 4\alpha n \|\nabla f(\boldsymbol{w}_k)\|_\infty.$$

Square on both sides,

$$\Delta_k^2 \leq 3\left(\frac{1}{4}\right)^{k-1} \Delta_1^2 + 48\alpha^2 H^2\varsigma^2 + 48\alpha^2 n^2 \|\nabla f(\boldsymbol{w}_k)\|_\infty^2.$$

We can apply the similar trick to Equation (6) and get

$$\Delta_1^2 \leq 8\alpha^2 n^2 \|\nabla f(\boldsymbol{w}_1)\|_\infty^2 + 8\alpha^2 n^2\varsigma^2.$$

This completes the proof of the second inequality in the lemma. Summing from $k = 1$ to $K$, we will get

$$\sum_{k=1}^{K} \Delta_k^2 = \Delta_1^2 + \sum_{k=2}^{K} \Delta_k^2$$

$$= \Delta_1^2 + 3\Delta_1^2 \sum_{k=2}^{K} \left(\frac{1}{4}\right)^{k-1} + 48\alpha^2 H^2\varsigma^2(K-1) + 48\alpha^2 n^2 \sum_{k=2}^{K} \|\nabla f(\boldsymbol{w}_k)\|_\infty^2$$

$$\leq \Delta_1^2 + 3\Delta_1^2 \sum_{k=1}^{\infty} \left(\frac{1}{4}\right)^{k} + 48\alpha^2 H^2\varsigma^2(K-1) + 48\alpha^2 n^2 \sum_{k=2}^{K} \|\nabla f(\boldsymbol{w}_k)\|_\infty^2$$

$$\leq 16\alpha^2 n^2 \|\nabla f(\boldsymbol{w}_1)\|_\infty^2 + 16\alpha^2 n^2\varsigma^2 + 48\alpha^2 H^2\varsigma^2(K-1) + 48\alpha^2 n^2 \sum_{k=2}^{K} \|\nabla f(\boldsymbol{w}_k)\|_\infty^2$$

$$\leq 16\alpha^2 n^2\varsigma^2 + 48\alpha^2 H^2\varsigma^2 K + 48\alpha^2 n^2 \sum_{k=1}^{K} \|\nabla f(\boldsymbol{w}_k)\|_\infty^2.$$

That completes the third inequality, and we have finished proving all three inequalities. $\qquad\square$

**Lemma 4.** *In Algorithm 4, if the learning rate $\alpha$ fulfills*

$$\alpha \leq \min\left\{\frac{1}{26(n+A)L_{2,\infty}}, \frac{1}{260nL_\infty}\right\},$$

*then the following inequalities hold:*

$$\sum_{k=1}^{K} \Delta_k^2 \leq 120\alpha^2 n^2\varsigma^2 + 64\alpha^2 A^2\varsigma^2 K + 48\alpha^2 n^2 \sum_{k=1}^{K} \|\nabla f(\boldsymbol{w}_k)\|_\infty^2.$$

*Proof.* With out the loss of generality, for all the $m \in \{2, \cdots, n+1\}$ and all the $k \in \{3, \cdots, K\}$,

$$\boldsymbol{w}_k^{(m)} = \boldsymbol{w}_k - \alpha \sum_{t=1}^{m-1} \nabla f\left(\boldsymbol{w}_k^{(t)}; \boldsymbol{x}_{\sigma_k(t)}\right)$$

$$= \boldsymbol{w}_k - \alpha \sum_{t=1}^{m-1} \nabla f\left(\boldsymbol{w}_{k-1}^{(\sigma_{k-1}^{-1}(\sigma_k(t)))}; \boldsymbol{x}_{\sigma_k(t)}\right)$$

$$- \alpha \sum_{t=1}^{m-1} \left(\nabla f\left(\boldsymbol{w}_k^{(t)}; \boldsymbol{x}_{\sigma_k(t)}\right) - \nabla f\left(\boldsymbol{w}_{k-1}^{(\sigma_{k-1}^{-1}(\sigma_k(t)))}; \boldsymbol{x}_{\sigma_k(t)}\right)\right).$$

Now add and subtract

$$\alpha \sum_{t=1}^{m-1} \frac{1}{n} \sum_{s=1}^{n} \nabla f\left(\boldsymbol{w}_{k-2}^{(s)}; \boldsymbol{x}_{\sigma_{k-2}(s)}\right) = \frac{\alpha(m-1)}{n} \sum_{t=1}^{n} \nabla f\left(\boldsymbol{w}_{k-2}^{(s)}; \boldsymbol{x}_{\sigma_{k-2}(s)}\right),$$

which gives

$$
\boldsymbol{w}_k^{(m)} = \boldsymbol{w}_k - \alpha \sum_{t=1}^{m-1} \left( \nabla f \left( \boldsymbol{w}_{k-1}^{(\sigma_{k-1}^{-1}(\sigma_k(t)))}; \boldsymbol{x}_{\sigma_k(t)} \right) - \frac{1}{n} \sum_{s=1}^{n} \nabla f \left( \boldsymbol{w}_{k-2}^{(s)}; \boldsymbol{x}_{\sigma_{k-2}(s)} \right) \right)
$$
$$
- \frac{\alpha(m-1)}{n} \sum_{t=1}^{n} \nabla f \left( \boldsymbol{w}_{k-2}^{(t)}; \boldsymbol{x}_{\sigma_{k-2}(t)} \right)
$$
$$
- \alpha \sum_{t=1}^{m-1} \left( \nabla f \left( \boldsymbol{w}_k^{(t)}; \boldsymbol{x}_{\sigma_k(t)} \right) - \nabla f \left( \boldsymbol{w}_{k-1}^{(\sigma_{k-1}^{-1}(\sigma_k(t)))}; \boldsymbol{x}_{\sigma_k(t)} \right) \right).
$$

We further add and subtract

$$
\frac{\alpha(m-1)}{n} \sum_{t=1}^{n} \nabla f(\boldsymbol{w}_{k-1}; \boldsymbol{x}_{\sigma_{k-2}(t)}) = \alpha(m-1)\nabla f(\boldsymbol{w}_{k-1})
$$

to arrive at

$$
\boldsymbol{w}_k^{(m)} = \boldsymbol{w}_k - \alpha \sum_{t=1}^{m-1} \left( \nabla f \left( \boldsymbol{w}_{k-1}^{(\sigma_{k-1}^{-1}(\sigma_k(t)))}; \boldsymbol{x}_{\sigma_k(t)} \right) - \frac{1}{n} \sum_{s=1}^{n} \nabla f \left( \boldsymbol{w}_{k-2}^{(s)}; \boldsymbol{x}_{\sigma_{k-2}(s)} \right) \right)
$$
$$
- \alpha(m-1)\nabla f(\boldsymbol{w}_{k-1}) + \frac{\alpha(m-1)}{n} \sum_{t=1}^{n} \left( \nabla f \left( \boldsymbol{w}_{k-1}; \boldsymbol{x}_{\sigma_{k-2}(t)} \right) - \nabla f \left( \boldsymbol{w}_{k-2}^{(t)}; \boldsymbol{x}_{\sigma_{k-2}(t)} \right) \right)
$$
$$
- \alpha \sum_{t=1}^{m-1} \left( \nabla f \left( \boldsymbol{w}_k^{(t)}; \boldsymbol{x}_{\sigma_k(t)} \right) - \nabla f \left( \boldsymbol{w}_{k-1}^{(\sigma_{k-1}^{-1}(\sigma_k(t)))}; \boldsymbol{x}_{\sigma_k(t)} \right) \right).
$$

We can now re-arrange, take norms on both sides and apply the triangle inequality,

$$
\left\| \boldsymbol{w}_k^{(m)} - \boldsymbol{w}_k \right\|_\infty \leq \alpha \left\| \sum_{t=1}^{m-1} \left( \nabla f \left( \boldsymbol{w}_{k-1}^{(\sigma_{k-1}^{-1}(\sigma_k(t)))}; \boldsymbol{x}_{\sigma_k(t)} \right) - \frac{1}{n} \sum_{s=1}^{n} \nabla f \left( \boldsymbol{w}_{k-2}^{(s)}; \boldsymbol{x}_{\sigma_{k-2}(s)} \right) \right) \right\|_\infty
$$
$$
+ \alpha(m-1)\|\nabla f(\boldsymbol{w}_{k-1})\|_\infty
$$
$$
+ \frac{\alpha(m-1)}{n} \left\| \sum_{t=1}^{n} \left( \nabla f \left( \boldsymbol{w}_{k-1}; \boldsymbol{x}_{\sigma_{k-2}(t)} \right) - \nabla f \left( \boldsymbol{w}_{k-2}^{(t)}; \boldsymbol{x}_{\sigma_{k-2}(t)} \right) \right) \right\|_\infty
$$
$$
+ \alpha \left\| \sum_{t=1}^{m-1} \left( \nabla f \left( \boldsymbol{w}_k^{(t)}; \boldsymbol{x}_{\sigma_k(t)} \right) - \nabla f \left( \boldsymbol{w}_{k-1}^{(\sigma_{k-1}^{-1}(\sigma_k(t)))}; \boldsymbol{x}_{\sigma_k(t)} \right) \right) \right\|_\infty.
$$

Similar to the proof of Lemma 3, for the last two terms, we simply apply the Assumption 1 to obtain

$$
\frac{\alpha(m-1)}{n} \left\| \sum_{t=1}^{n} \left( \nabla f \left( \boldsymbol{w}_{k-1}; \boldsymbol{x}_{\sigma_{k-2}(t)} \right) - \nabla f \left( \boldsymbol{w}_{k-2}^{(t)}; \boldsymbol{x}_{\sigma_{k-2}(t)} \right) \right) \right\|_\infty
$$
$$
\leq \frac{\alpha(m-1)}{n} \sum_{t=1}^{n} \left\| \nabla f \left( \boldsymbol{w}_{k-1}; \boldsymbol{x}_{\sigma_{k-2}(t)} \right) - \nabla f \left( \boldsymbol{w}_{k-2}^{(t)}; \boldsymbol{x}_{\sigma_{k-2}(t)} \right) \right\|_\infty
$$
$$
\leq \frac{\alpha(m-1)L_\infty}{n} \sum_{t=1}^{n} \left\| \boldsymbol{w}_{k-1} - \boldsymbol{w}_{k-2}^{(t)} \right\|_\infty
$$
$$
\leq \frac{\alpha(m-1)L_\infty}{n} \sum_{t=1}^{n} \left( \|\boldsymbol{w}_{k-1} - \boldsymbol{w}_{k-2}\|_\infty + \left\| \boldsymbol{w}_{k-2} - \boldsymbol{w}_{k-2}^{(t)} \right\|_\infty \right)
$$
$$
\leq \alpha(m-1)L_\infty(2\Delta_{k-2}),
$$

and

$$
\alpha \left\| \sum_{t=1}^{m-1} \left( \nabla f \left( \boldsymbol{w}_k^{(t)}; \boldsymbol{x}_{\sigma_k(t)} \right) - \nabla f \left( \boldsymbol{w}_{k-1}^{(\sigma_{k-1}^{-1}(\sigma_k(t)))}; \boldsymbol{x}_{\sigma_k(t)} \right) \right) \right\|_\infty
$$

$$\leq \alpha L_\infty \sum_{t=1}^{m-1} \left\| \boldsymbol{w}_k^{(t)} - \boldsymbol{w}_{k-1}^{(\sigma_{k-1}^{-1}(\sigma_k(t)))} \right\|_\infty$$

$$\leq \alpha L_\infty \sum_{t=1}^{m-1} \left( \left\| \boldsymbol{w}_k^{(t)} - \boldsymbol{w}_k \right\|_\infty + \left\| \boldsymbol{w}_k - \boldsymbol{w}_{k-1} \right\|_\infty + \left\| \boldsymbol{w}_{k-1} - \boldsymbol{w}_{k-1}^{(\sigma_{k-1}^{-1}(\sigma_k(t)))} \right\|_\infty \right)$$

$$\leq \alpha(m-1) L_\infty (\Delta_k + 2\Delta_{k-1}).$$

For brevity, we use $r_k$, $k \geq 3$ to denote

$$r_k = \max_{2 \leq m \leq n+1} \left\| \sum_{t=1}^{m-1} \left( \nabla f\left( \boldsymbol{w}_{k-1}^{(\sigma_{k-1}^{-1}(\sigma_k(t)))}; \boldsymbol{x}_{\sigma_k(t)} \right) - \frac{1}{n} \sum_{s=1}^{n} \nabla f\left( \boldsymbol{w}_{k-2}^{(s)}; \boldsymbol{x}_{\sigma_{k-2}(s)} \right) \right) \right\|_\infty.$$

Then the expression of $\Delta_k$ can be written as

$$\Delta_k \leq \alpha r_k + \alpha n L_\infty \Delta_k + 2\alpha n L_\infty \Delta_{k-1} + 2\alpha n L_\infty \Delta_{k-2} + \alpha n \|\nabla f(\boldsymbol{w}_{k-1})\|_\infty.$$

Now square on both sides, it gives us

$$\Delta_k^2 \leq 5\alpha^2 r_k^2 + 5\alpha^2 n^2 L_\infty^2 \Delta_k^2 + 20\alpha^2 n^2 L_\infty^2 \Delta_{k-1}^2 + 20\alpha^2 n^2 L_\infty^2 \Delta_{k-2}^2 + 5\alpha^2 n^2 \|\nabla f(\boldsymbol{w}_{k-1})\|_\infty^2.$$

Now summing $k$ from $k=3$ to $K$ on both sides of the inequality, we get

$$\sum_{k=3}^{K} \Delta_k^2$$

$$\leq 5\alpha^2 \sum_{k=3}^{K} r_k^2 + 5\alpha^2 n^2 L_\infty^2 \sum_{k=3}^{K} \Delta_k^2 + 20\alpha^2 n^2 L_\infty^2 \sum_{k=3}^{K} \Delta_{k-1}^2 + 20\alpha^2 n^2 L_\infty^2 \sum_{k=3}^{K} \Delta_{k-2}^2 + 5\alpha^2 n^2 \sum_{k=3}^{K} \|\nabla f(\boldsymbol{w}_{k-1})\|_\infty^2$$

$$\leq 5\alpha^2 \sum_{k=3}^{K} r_k^2 + 45\alpha^2 n^2 L_\infty^2 \sum_{k=3}^{K} \Delta_k^2 + 40\alpha^2 n^2 L_\infty^2 \Delta_2^2 + 20\alpha^2 n^2 L_\infty^2 \Delta_1^2 + 5\alpha^2 n^2 \sum_{k=2}^{K-1} \|\nabla f(\boldsymbol{w}_k)\|_\infty^2.$$

Now we apply Lemma 6 to replace the first term,

$$\sum_{k=3}^{K} \Delta_k^2 \leq 30\alpha^2 n^2 \varsigma^2 + 30\alpha^2 A^2 \varsigma^2 (K-2) + 5\alpha^2 \cdot (792 n^2 L_\infty^2 + 78 A^2 L_{2,\infty}^2) \sum_{k=3}^{K} \Delta_k^2$$

$$+ 5\alpha^2 \cdot (24 n^2 L_{2,\infty} + 24 A^2 L_{2,\infty}^2 + 384 n^2 L_\infty^2) \Delta_1^2$$

$$+ 5\alpha^2 \cdot (6 n^2 L_{2,\infty}^2 + 78 A^2 L_{2,\infty}^2 + 768 n^2 L_\infty^2) \Delta_2^2.$$

$$+ 45\alpha^2 n^2 L_\infty^2 \sum_{k=3}^{K} \Delta_k^2 + 40\alpha^2 n^2 L_\infty^2 \Delta_2^2 + 20\alpha^2 n^2 L_\infty^2 \Delta_1^2 + 5\alpha^2 n^2 \sum_{k=2}^{K-1} \|\nabla f(\boldsymbol{w}_k)\|_\infty^2$$

$$= 30\alpha^2 n^2 \varsigma^2 + 30\alpha^2 A^2 \varsigma^2 (K-2) + 5\alpha^2 \cdot (781 n^2 L_\infty^2 + 78 A^2 L_{2,\infty}^2) \sum_{k=3}^{K} \Delta_k^2$$

$$+ 5\alpha^2 \cdot (24 n^2 L_{2,\infty} + 24 A^2 L_{2,\infty}^2 + 388 n^2 L_\infty^2) \Delta_1^2$$

$$+ 5\alpha^2 \cdot (6 n^2 L_{2,\infty}^2 + 78 A^2 L_{2,\infty}^2 + 776 n^2 L_\infty^2) \Delta_2^2.$$

$$+ 5\alpha^2 n^2 \sum_{k=2}^{K-1} \|\nabla f(\boldsymbol{w}_k)\|_\infty^2$$

$$\leq 30\alpha^2 n^2 \varsigma^2 + 30\alpha^2 A^2 \varsigma^2 (K-2) + \Delta_1^2 + \Delta_2^2 + \frac{1}{2} \sum_{k=3}^{K} \Delta_k^2 + 5\alpha^2 n^2 \sum_{k=2}^{K-1} \|\nabla f(\boldsymbol{w}_k)\|_\infty^2,$$

where in the last step, we apply the learning rate requirement that

$$\alpha \leq \min\left\{ \frac{1}{26(n+A)L_{2,\infty}}, \frac{1}{260 n L_\infty} \right\}.$$

Now solve the LHS,

$$\sum_{k=3}^{K} \Delta_k^2 \leq 60\alpha^2 n^2 \varsigma^2 + 60\alpha^2 A^2 \varsigma^2 (K-2) + 2\Delta_1^2 + 2\Delta_2^2 + 10\alpha^2 n^2 \sum_{k=2}^{K-1} \|\nabla f(\boldsymbol{w}_k)\|_\infty^2$$

We have now proved $k \geq 3$. We next discuss the $k = 1, 2$ case, note that these cases follow exactly Equation (6), so that we can reapply the $k = 1$ case from Lemma 3 and obtain

$$\Delta_k^2 \leq 8\alpha^2 n^2 \|\nabla f(\boldsymbol{w}_k)\|_\infty^2 + 8\alpha^2 n^2 \varsigma^2, k = 1, 2.$$

Now we can sum over all the epochs as follows

$$\sum_{k=1}^{K} \Delta_k^2 = \sum_{k=1}^{2} \Delta_k^2 + \sum_{k=3}^{K} \Delta_k^2$$

$$\leq 24\alpha^2 n^2 \sum_{k=1}^{2} \|\nabla f(\boldsymbol{w}_k)\|_\infty^2 + 48\alpha^2 n^2 \varsigma^2 + 60\alpha^2 n^2 \varsigma^2 + 60\alpha^2 A^2 \varsigma^2 (K-2) + 10\alpha^2 n^2 \sum_{k=2}^{K-1} \|\nabla f(\boldsymbol{w}_k)\|_\infty^2$$

$$\leq 120\alpha^2 n^2 \varsigma^2 + 64\alpha^2 A^2 \varsigma^2 K + 48\alpha^2 n^2 \sum_{k=1}^{K} \|\nabla f(\boldsymbol{w}_k)\|_\infty^2.$$

That completes the proof. $\qquad\square$

**Lemma 5.** *Consider a group of time-variant input vectors $\boldsymbol{z}_{1,k}, \ldots, \boldsymbol{z}_{n,k} \in \mathbb{R}^d$ changing over time $k = 3, \cdots, K$ that satisfy the following conditions:*

$$\|\boldsymbol{z}_{i,k+1} - \boldsymbol{z}_{i,k}\|_\infty \leq a_k, \forall k, i$$

$$\left\|\sum_{i=1}^{n} \boldsymbol{z}_{i,k}\right\|_\infty \leq b_k, \forall k$$

$$\|\boldsymbol{z}_{i,k}\|_2 \leq c_k, \forall k, i.$$

*Now considering a sequence of ordering of $\{\sigma_k\}_{k=3}^{K}$ that fulfills condition: for any $k \geq 3$, $\sigma_k$ and $\sigma_{k+1}$ are the input and output of Algorithm 3, then it holds that ,*

$$\sum_{k=3}^{K} \max_{1 \leq t \leq n} \left\|\sum_{j=1}^{t} \boldsymbol{z}_{\sigma_k(j),k}\right\|_\infty^2 \leq 2 \max_{1 \leq t \leq n} \left\|\sum_{j=1}^{t} \boldsymbol{z}_{\sigma_3(j),3}\right\|_\infty^2 + \sum_{k=3}^{K} \left(Ac_k + 2b_k + 2a_k n\right)^2.$$

*Proof.* For convenience, we use the notation $\tilde{r}_k$ to represent

$$\tilde{r}_k = \max_{1 \leq t \leq n} \left\|\sum_{j=1}^{t} \boldsymbol{z}_{\sigma_k(j),k}\right\|_\infty.$$

For any $k \geq 3$, by the Triangle Inequality we have

$$\max_{1 \leq t \leq n} \left\|\sum_{j=1}^{t} \boldsymbol{z}_{\sigma_{k+1}(j),k+1}\right\|_\infty = \max_{1 \leq t \leq n} \left\|\sum_{j=1}^{t} \boldsymbol{z}_{\sigma_{k+1}(j),k} + \boldsymbol{z}_{\sigma_{k+1}(j),k+1} - \boldsymbol{z}_{\sigma_{k+1}(j),k}\right\|_\infty$$

$$\leq \max_{1 \leq t \leq n} \left[\left\|\sum_{j=1}^{t} \boldsymbol{z}_{\sigma_{k+1}(j),k}\right\|_\infty + \left\|\sum_{j=1}^{t} \left(\boldsymbol{z}_{\sigma_{k+1}(j),k+1} - \boldsymbol{z}_{\sigma_{k+1}(j),k}\right)\right\|_\infty\right]$$

$$\leq \max_{1 \leq t \leq n} \left\|\sum_{j=1}^{t} \boldsymbol{z}_{\sigma_{k+1}(j),k}\right\|_\infty + \max_{1 \leq t \leq n} \left\|\sum_{j=1}^{t} \left(\boldsymbol{z}_{\sigma_{k+1}(j),k+1} - \boldsymbol{z}_{\sigma_{k+1}(j),k}\right)\right\|_\infty$$

$$\leq \max_{1 \leq t \leq n} \left\|\sum_{j=1}^{t} \boldsymbol{z}_{\sigma_{k+1}(j),k}\right\|_\infty + \max_{1 \leq t \leq n} \sum_{j=1}^{t} \|\boldsymbol{z}_{\sigma_{k+1}(j),k+1} - \boldsymbol{z}_{\sigma_{k+1}(j),k}\|_\infty$$

$$\leq \max_{1 \leq t \leq n} \left\| \sum_{j=1}^{t} \boldsymbol{z}_{\sigma_{k+1}(j),k} \right\|_{\infty} + a_k n.$$

We now derive the relation between $\sigma_{k+1}$ and $\sigma_k$. Suppose we run Algorithm 3 and obtain the signs for all the vectors. Denote $\epsilon_{k,j}$ as the sign assigned to $\boldsymbol{z}_{j,k}$. Let all the vector indices with positive signs form a set $M^+$ and indices with negative signs form a set $M^-$. Then we know that for any $1 \leq t \leq n$,

$$\sum_{j=1}^{t} \boldsymbol{z}_{\sigma_k(j),k} + \sum_{j=1}^{t} \epsilon_{k,j} \boldsymbol{z}_{\sigma_k(j),k} = 2 \cdot \sum_{j \in M^+ \cap [k]} \boldsymbol{z}_{\sigma_k(j),k}$$

$$\sum_{j=1}^{t} \boldsymbol{z}_{\sigma_k(j),k} - \sum_{j=1}^{t} \epsilon_{k,j} \boldsymbol{z}_{\sigma_k(j),k} = 2 \cdot \sum_{j \in M^- \cap [k]} \boldsymbol{z}_{\sigma_k(j),k}.$$

By using the triangle inequality, for any $k$

$$\left\| \sum_{j \in M^+ \cap [k]} \boldsymbol{z}_{\sigma_k(j),k} \right\|_{\infty} \leq \frac{Ac_k + \tilde{r}_k}{2}$$

$$\left\| \sum_{j \in M^- \cap [k]} \boldsymbol{z}_{\sigma_k(j),k} \right\|_{\infty} \leq \frac{Ac_k + \tilde{r}_k}{2}.$$

Given these bounds, now we can consider the permutation of $\sigma_{k+1}$, suppose when $t = t_0$, the partial sum of $\sigma_{k+1}$ reaches it maximum, then if $t_0 \leq |M^+|$, from the previous bound, we obtain

$$\left\| \sum_{j=1}^{t_0} \boldsymbol{z}_{\sigma_{k+1}(j),k} \right\|_{\infty} \leq \frac{Ac_k + \tilde{r}_k}{2} \leq b_k + \frac{Ac_k + \tilde{r}_k}{2}.$$

On the other hand, if the $t_0 > |M^+|$, and we obtain

$$\left\| \sum_{j=1}^{t_0} \boldsymbol{z}_{\sigma_{k+1}(j),k} \right\|_{\infty} = \left\| \sum_{j=1}^{n} \boldsymbol{z}_{\sigma_{k+1}(j),k} - \sum_{j=t_0+1}^{n} \boldsymbol{z}_{\sigma_{k+1}(j),k} \right\|_{\infty}$$

$$\leq \left\| \sum_{j=1}^{n} \boldsymbol{z}_{\sigma_{k+1}(j),k} \right\|_{\infty} + \left\| \sum_{j=t_0+1}^{n} \boldsymbol{z}_{\sigma_{k+1}(j),k} \right\|_{\infty}$$

$$\leq b_k + \left\| \sum_{j \in M^- \cap [k]} \boldsymbol{z}_{\sigma_k(j),k} \right\|_{\infty}$$

$$\leq b_k + \frac{Ac_k + \tilde{r}_k}{2}.$$

And so that,

$$\tilde{r}_{k+1} = \max_{1 \leq t \leq n} \left\| \sum_{j=1}^{t} \boldsymbol{z}_{\sigma_{k+1}(j),k+1} \right\|_{\infty} \leq \max_{1 \leq t \leq n} \left\| \sum_{j=1}^{t} \boldsymbol{z}_{\sigma_{k+1}(j),k} \right\|_{\infty} + an$$

$$\leq \frac{1}{2} \tilde{r}_k + \frac{1}{2} \left( Ac_k + 2b_k + 2a_k n \right).$$

Square on both sides, we get

$$\tilde{r}_{k+1}^2 \leq \frac{1}{2} \tilde{r}_k^2 + \frac{1}{2} \left( Ac_k + 2b_k + 2a_k n \right)^2.$$

Then we sum from $k = 3$ to $K - 1$, we get

$$\tilde{r}_3^2 + \sum_{k=3}^{K-1} \tilde{r}_{k+1}^2 \leq \tilde{r}_3^2 + \frac{1}{2} \sum_{k=3}^{K-1} \tilde{r}_k^2 + \frac{1}{2} \sum_{k=3}^{K-1} \left( Ac_k + 2b_k + 2a_k n \right)^2,$$

which implies

$$\sum_{k=3}^{K} \tilde{r}_k^2 \le \tilde{r}_3^2 + \frac{1}{2}\sum_{k=3}^{K}\tilde{r}_k^2 + \frac{1}{2}\sum_{k=3}^{K}(Ac_k + 2b_k + 2a_k n)^2,$$

moving the terms and we finally get

$$\sum_{k=3}^{K}\tilde{r}_k^2 \le 2\tilde{r}_3^2 + \sum_{k=3}^{K}(Ac_k + 2b_k + 2a_k n)^2.$$

Replace the $\tilde{r}_k$, we finally get

$$\sum_{k=3}^{K}\max_{1\le t\le n}\left\|\sum_{j=1}^{t}\boldsymbol{z}_{\sigma_k(j),k}\right\|_\infty^2 \le 2\max_{1\le t\le n}\left\|\sum_{j=1}^{t}\boldsymbol{z}_{\sigma_3(j),3}\right\|_\infty^2 + \sum_{k=3}^{K}(Ac_k + 2b_k + 2a_k n)^2.$$

That completes the proof. $\qquad\square$

**Lemma 6.** *In Algorithm 4, for $k \ge 3$, with notation of*

$$r_k = \max_{1\le t\le n}\left\|\sum_{j=1}^{t}\left(\nabla f\left(\boldsymbol{w}_{k-1}^{(\sigma_{k-1}^{-1}(\sigma_k(j)))};\boldsymbol{x}_{\sigma_k(j)}\right) - \frac{1}{n}\sum_{s=0}^{n-1}\nabla f\left(\boldsymbol{w}_{k-2}^{(s)};\boldsymbol{x}_{\sigma_{k-2}(s)}\right)\right)\right\|_\infty,$$

*we have*

$$\begin{aligned}
\sum_{k=3}^{K} r_k^2 \le\, & 6n^2\varsigma^2 + 6A^2\varsigma^2(K-2) + (792n^2 L_\infty^2 + 78A^2 L_{2,\infty}^2)\sum_{k=3}^{K}\Delta_k^2 \\
& + (24n^2 L_{2,\infty} + 24A^2 L_{2,\infty}^2 + 384n^2 L_\infty^2)\Delta_1^2 \\
& + (6n^2 L_{2,\infty}^2 + 78A^2 L_{2,\infty}^2 + 768n^2 L_\infty^2)\Delta_2^2.
\end{aligned}$$

*Proof.* We will apply Lemma 5 to the main derivation. Denote

$$\boldsymbol{z}_{j,k} = \nabla f\left(\boldsymbol{w}_{k-1}^{(\sigma_{k-1}^{-1}(j))};\boldsymbol{x}_j\right) - \frac{1}{n}\sum_{s=1}^{n}\nabla f\left(\boldsymbol{w}_{k-2}^{(s)};\boldsymbol{x}_{\sigma_{k-2}(s)}\right),$$

and so

$$\boldsymbol{z}_{\sigma_k(j),k} = \nabla f\left(\boldsymbol{w}_{k-1}^{(\sigma_{k-1}^{-1}(\sigma_k(j)))};\boldsymbol{x}_{\sigma_k(j)}\right) - \frac{1}{n}\sum_{s=1}^{n}\nabla f\left(\boldsymbol{w}_{k-2}^{(s)};\boldsymbol{x}_{\sigma_{k-2}(s)}\right).$$

We first need to derive $a_k$, $b_k$, $c_k$ in Lemma 5 in order to apply it, we will repeatedly use Assumption 1. In the derivation, we will also use the relation that $\|\boldsymbol{a}\|_\infty \le \|\boldsymbol{a}\|_2$.

First, we derive $a_k$,

$$\begin{aligned}
& \|\boldsymbol{z}_{j,k+1} - \boldsymbol{z}_{j,k}\|_\infty \\
\le & \left\|\nabla f\left(\boldsymbol{w}_k^{(\sigma_k^{-1}(j))};\boldsymbol{x}_j\right) - \nabla f\left(\boldsymbol{w}_{k-1}^{(\sigma_{k-1}^{-1}(j))};\boldsymbol{x}_j\right)\right\|_\infty \\
& + \left\|\frac{1}{n}\sum_{s=1}^{n}\nabla f\left(\boldsymbol{w}_{k-1}^{(s)};\boldsymbol{x}_{\sigma_{k-1}(s)}\right) - \frac{1}{n}\sum_{s=1}^{n}\nabla f\left(\boldsymbol{w}_{k-2}^{(s)};\boldsymbol{x}_{\sigma_{k-2}(s)}\right)\right\|_\infty \\
= & \left\|\nabla f\left(\boldsymbol{w}_k^{(\sigma_k^{-1}(j))};\boldsymbol{x}_j\right) - \nabla f\left(\boldsymbol{w}_{k-1}^{(\sigma_{k-1}^{-1}(j))};\boldsymbol{x}_j\right)\right\|_\infty \\
& + \left\|\frac{1}{n}\sum_{s=1}^{n}\nabla f\left(\boldsymbol{w}_{k-1}^{\sigma_{k-1}^{-1}(s)};\boldsymbol{x}_s\right) - \frac{1}{n}\sum_{s=1}^{n}\nabla f\left(\boldsymbol{w}_{k-2}^{\sigma_{k-2}^{-1}(s)};\boldsymbol{x}_s\right)\right\|_\infty
\end{aligned}$$

$$\leq L_\infty \left\| \boldsymbol{w}_k^{(\sigma_k^{-1}(j))} - \boldsymbol{w}_{k-1}^{(\sigma_{k-1}^{-1}(j))} \right\|_\infty + \frac{L_\infty}{n} \sum_{s=1}^n \left\| \boldsymbol{w}_{k-1}^{\sigma_{k-1}^{-1}(s)} - \boldsymbol{w}_{k-2}^{\sigma_{k-2}^{-1}(s)} \right\|_\infty$$

$$\leq L_\infty \left( \left\| \boldsymbol{w}_k^{(\sigma_k^{-1}(j))} - \boldsymbol{w}_k \right\|_\infty + \left\| \boldsymbol{w}_k - \boldsymbol{w}_{k-1} \right\|_\infty + \left\| \boldsymbol{w}_{k-1} - \boldsymbol{w}_{k-1}^{(\sigma_{k-1}^{-1}(j))} \right\|_\infty \right)$$

$$+ \frac{L_\infty}{n} \sum_{s=1}^n \left( \left\| \boldsymbol{w}_{k-1}^{\sigma_{k-1}^{-1}(s)} - \boldsymbol{w}_{k-1} \right\|_\infty + \left\| \boldsymbol{w}_{k-1} - \boldsymbol{w}_{k-2} \right\|_\infty + \left\| \boldsymbol{w}_{k-2} - \boldsymbol{w}_{k-2}^{\sigma_{k-2}^{-1}(s)} \right\|_\infty \right)$$

$$\leq L_\infty \left( \Delta_k + 2\Delta_{k-1} \right) + L_\infty \left( \Delta_{k-1} + 2\Delta_{k-2} \right)$$

$$= L_\infty \left( \Delta_k + 3\Delta_{k-1} + 2\Delta_{k-2} \right).$$

We proceed to derive $b_k$,

$$\left\| \sum_{j=1}^n \boldsymbol{z}_{j,k} \right\|_\infty = \left\| \sum_{j=1}^n \left( \nabla f \left( \boldsymbol{w}_{k-1}^{(\sigma_{k-1}^{-1}(j))}; \boldsymbol{x}_j \right) - \frac{1}{n} \sum_{s=1}^n \nabla f \left( \boldsymbol{w}_{k-2}^{\sigma_{k-2}^{-1}(s)}; \boldsymbol{x}_s \right) \right) \right\|_\infty$$

$$\leq \left\| \sum_{j=1}^n \left( \nabla f \left( \boldsymbol{w}_{k-1}^{(\sigma_{k-1}^{-1}(j))}; \boldsymbol{x}_j \right) - \frac{1}{n} \sum_{s=1}^n \nabla f \left( \boldsymbol{w}_{k-1}^{\sigma_{k-1}^{-1}(s)}; \boldsymbol{x}_s \right) \right) \right\|_\infty$$

$$+ \left\| \sum_{j=1}^n \left( \frac{1}{n} \sum_{s=1}^n \nabla f \left( \boldsymbol{w}_{k-1}^{\sigma_{k-1}^{-1}(s)}; \boldsymbol{x}_s \right) - \frac{1}{n} \sum_{s=1}^n \nabla f \left( \boldsymbol{w}_{k-2}^{\sigma_{k-2}^{-1}(s)}; \boldsymbol{x}_s \right) \right) \right\|_\infty$$

$$\leq 0 + \sum_{j=1}^n \left\| \frac{1}{n} \sum_{s=1}^n \nabla f \left( \boldsymbol{w}_{k-1}^{\sigma_{k-1}^{-1}(s)}; \boldsymbol{x}_s \right) - \frac{1}{n} \sum_{s=1}^n \nabla f \left( \boldsymbol{w}_{k-2}^{\sigma_{k-2}^{-1}(s)}; \boldsymbol{x}_s \right) \right\|_\infty$$

$$\leq \sum_{j=1}^n \frac{1}{n} \sum_{s=1}^n L_\infty \left\| \boldsymbol{w}_{k-1}^{\sigma_{k-1}^{-1}(s)} - \boldsymbol{w}_{k-2}^{\sigma_{k-2}^{-1}(s)} \right\|_\infty$$

$$\leq \sum_{j=1}^n \frac{1}{n} \sum_{s=1}^n L_\infty \left( \left\| \boldsymbol{w}_{k-1}^{\sigma_{k-1}^{-1}(s)} - \boldsymbol{w}_{k-1} \right\|_\infty + \left\| \boldsymbol{w}_{k-1} - \boldsymbol{w}_{k-2} \right\|_\infty + \left\| \boldsymbol{w}_{k-2} - \boldsymbol{w}_{k-2}^{\sigma_{k-2}^{-1}(s)} \right\|_\infty \right)$$

$$\leq n L_\infty (\Delta_{k-1} + 2\Delta_{k-2}).$$

And finally for $c_k$,

$$\| \boldsymbol{z}_{j,k} \|_2 = \left\| \nabla f \left( \boldsymbol{w}_{k-1}^{(\sigma_{k-1}^{-1}(j))}; \boldsymbol{x}_j \right) - \frac{1}{n} \sum_{s=1}^n \nabla f \left( \boldsymbol{w}_{k-2}^{\sigma_{k-2}^{-1}(s)}; \boldsymbol{x}_s \right) \right\|_2$$

$$\leq \left\| \nabla f \left( \boldsymbol{w}_{k-1}^{(\sigma_{k-1}^{-1}(j))}; \boldsymbol{x}_j \right) - \frac{1}{n} \sum_{s=1}^n \nabla f \left( \boldsymbol{w}_{k-1}^{\sigma_{k-1}^{-1}(j)}; \boldsymbol{x}_s \right) \right\|_2$$

$$+ \left\| \frac{1}{n} \sum_{s=1}^n \nabla f \left( \boldsymbol{w}_{k-1}^{\sigma_{k-1}^{-1}(j)}; \boldsymbol{x}_s \right) - \frac{1}{n} \sum_{s=1}^n \nabla f \left( \boldsymbol{w}_{k-1}; \boldsymbol{x}_s \right) \right\|_2$$

$$+ \left\| \frac{1}{n} \sum_{s=1}^n \nabla f \left( \boldsymbol{w}_{k-1}; \boldsymbol{x}_s \right) - \frac{1}{n} \sum_{s=1}^n \nabla f \left( \boldsymbol{w}_{k-1}^{\sigma_{k-1}^{-1}(s)}; \boldsymbol{x}_s \right) \right\|_2$$

$$+ \left\| \frac{1}{n} \sum_{s=1}^n \nabla f \left( \boldsymbol{w}_{k-1}^{\sigma_{k-1}^{-1}(s)}; \boldsymbol{x}_s \right) - \frac{1}{n} \sum_{s=1}^n \nabla f \left( \boldsymbol{w}_{k-2}^{\sigma_{k-2}^{-1}(s)}; \boldsymbol{x}_s \right) \right\|_2$$

$$\leq \varsigma + 2L_{2,\infty} \Delta_{k-1} + L_{2,\infty} \left( \left\| \boldsymbol{w}_{k-1}^{\sigma_{k-1}^{-1}(s)} - \boldsymbol{w}_{k-1} \right\|_\infty + \left\| \boldsymbol{w}_{k-1} - \boldsymbol{w}_{k-2} \right\|_\infty + \left\| \boldsymbol{w}_{k-2} - \boldsymbol{w}_{k-2}^{\sigma_{k-2}^{-1}(s)} \right\|_\infty \right)$$

$$\leq \varsigma + L_{2,\infty} (3\Delta_{k-1} + 2\Delta_{k-2}).$$

Now we have the $a_k$, $b_k$ and $c_k$ ready, we can now apply Lemma 5, and get

$$\sum_{k=3}^K r_k^2 \leq 2 r_3^2 + \sum_{k=3}^K \left[ A\varsigma + 2n L_\infty \Delta_k + (3A L_{2,\infty} + 8n L_\infty) \Delta_{k-1} + (2A L_{2,\infty} + 8n L_\infty) \Delta_{k-2} \right]^2.$$

Finally, we look at $r_3$, note that

$$r_3 = \max_{1 \le t \le n} \left\| \sum_{j=1}^{t} \left( \nabla f\left(\boldsymbol{w}_2^{(\sigma_2^{-1}(\sigma_3(j)))}; \boldsymbol{x}_{\sigma_3(j)}\right) - \frac{1}{n}\sum_{s=0}^{n-1} \nabla f\left(\boldsymbol{w}_1^{(s)}; \boldsymbol{x}_{\sigma_1(s)}\right) \right) \right\|_{\infty}$$

$$\le \sum_{j=1}^{n} \left\| \nabla f\left(\boldsymbol{w}_2^{(\sigma_2^{-1}(\sigma_3(j)))}; \boldsymbol{x}_{\sigma_3(j)}\right) - \frac{1}{n}\sum_{s=0}^{n-1} \nabla f\left(\boldsymbol{w}_1^{(s)}; \boldsymbol{x}_{\sigma_1(s)}\right) \right\|_{\infty}$$

$$\le c_3 n$$

$$\le n\varsigma + nL_{2,\infty}(3\Delta_2 + 2\Delta_1).$$

Square on both sides, we can get

$$r_3^2 \le 3n^2\varsigma^2 + 27n^2 L_{2,\infty}^2 \Delta_2^2 + 12n^2 L_{2,\infty}^2 \Delta_1^2.$$

Push it back we can finally obtain the bound as

$$\sum_{k=3}^{K} r_k^2 \le 6n^2\varsigma^2 + 6n^2 L_{2,\infty}^2 \Delta_2^2 + 24n^2 L_{2,\infty}^2 \Delta_1^2$$

$$+ \sum_{k=3}^{K} \left[ A\varsigma + 2nL_\infty \Delta_k + (3AL_{2,\infty} + 8nL_\infty)\Delta_{k-1} + (2AL_{2,\infty} + 8nL_\infty)\Delta_{k-2} \right]^2$$

$$\le 6n^2\varsigma^2 + 6n^2 L_{2,\infty}^2 \Delta_2^2 + 24n^2 L_{2,\infty}^2 \Delta_1^2$$

$$+ 6A^2\varsigma^2(K-2) + 24n^2 L_\infty^2 \sum_{k=3}^{K} \Delta_k^2 + 54A^2 L_{2,\infty}^2 \sum_{k=3}^{K} \Delta_{k-1}^2 + 384n^2 L_\infty^2 \sum_{k=3}^{K} \Delta_{k-1}^2$$

$$+ 24A^2 L_{2,\infty}^2 \sum_{k=3}^{K} \Delta_{k-2}^2 + 384n^2 L_\infty^2 \sum_{k=3}^{K} \Delta_{k-2}^2$$

$$\le 6n^2\varsigma^2 + 6A^2\varsigma^2(K-2) + (792n^2 L_\infty^2 + 78A^2 L_{2,\infty}^2) \sum_{k=3}^{K} \Delta_k^2$$

$$+ (24n^2 L_{2,\infty} + 24A^2 L_{2,\infty}^2 + 384n^2 L_\infty^2)\Delta_1^2$$

$$+ (6n^2 L_{2,\infty}^2 + 78A^2 L_{2,\infty}^2 + 768n^2 L_\infty^2)\Delta_2^2.$$

That completes the proof. $\qquad\qquad\qquad\qquad\qquad\qquad\qquad\qquad\qquad\qquad\qquad\qquad\qquad\square$

**Lemma 7.** *In Algorithm 4, under the PL condition, it holds that*

$$\sum_{k=1}^{K-1} \rho^{K-1-k}\Delta_k^2 \le 960\alpha^2\rho^K n^2\varsigma^2 + \frac{120\alpha^2 A^2\varsigma^2}{1-\rho} + 64\alpha^2 n^2 \sum_{k=1}^{K-1} \rho^{K-1-k}\|\nabla f(\boldsymbol{w}_k)\|_\infty^2.$$

*Proof.* This lemma proves a weighted summation sequence for PL case convergence of GraB. Many of the steps are overlapping with Lemma 5 and Lemma 6. For brevity, we will omit some of the steps, while their detailed derivations can be found in the proof of Lemma 5 and Lemma 6.

First we define $r_k$ for any $k \ge 3$ the same as Lemma 6

$$r_k = \max_{1 \le t \le n} \left\| \sum_{j=1}^{t} \left( \nabla f\left(\boldsymbol{w}_{k-1}^{(\sigma_{k-1}^{-1}(\sigma_k(j)))}; \boldsymbol{x}_{\sigma_k(j)}\right) - \frac{1}{n}\sum_{s=0}^{n-1} \nabla f\left(\boldsymbol{w}_{k-2}^{(s)}; \boldsymbol{x}_{\sigma_{k-2}(s)}\right) \right) \right\|_{\infty}.$$

From Lemma 5 and Lemma 6 we know that for any $k \ge 3$,

$$r_{k+1}^2 \le \frac{1}{2}r_k^2 + \frac{1}{2}\left(Ac_k + 2b_k + 2a_k n\right)^2,$$

where the definition of $a_k$, $b_k$ and $c_k$ can be found in Lemma 6. We times $\rho^{K-1-(k+1)}$ on both sides and obtain

$$\rho^{K-1-(k+1)}r_{k+1}^2 \le \frac{1}{2\rho}\rho^{K-1-k}r_k^2 + \frac{1}{2\rho}\rho^{K-1-k}\left(Ac_k + 2b_k + 2a_k n\right)^2.$$

Summing from $k = 3$ to $K - 2$, we obtain

$$\rho^{K-1-(3)}r_3^2 + \sum_{k=3}^{K-2} \rho^{K-1-(k+1)}r_{k+1}^2 \leq \rho^{K-1-(3)}r_3^2 + \frac{1}{2\rho}\sum_{k=3}^{K-2}\rho^{K-1-k}r_k^2 + \frac{1}{2\rho}\sum_{k=3}^{K-2}\rho^{K-1-k}\left(Ac_k + 2b_k + 2a_k n\right)^2.$$

This implies that

$$\left(1 - \frac{1}{2\rho}\right)\sum_{k=3}^{K-1}\rho^{K-1-k}r_k^2 \leq \rho^{K-4}r_3^2 + \frac{1}{2\rho}\sum_{k=3}^{K-1}\rho^{K-1-k}\left(Ac_k + 2b_k + 2a_k n\right)^2.$$

We times $2\rho$ on both sides, it gives us

$$(2\rho - 1)\sum_{k=3}^{K-1}\rho^{K-1-k}r_k^2 \leq 2\rho^{K-3}r_3^2 + \sum_{k=3}^{K-1}\rho^{K-1-k}\left(Ac_k + 2b_k + 2a_k n\right)^2.$$

Note that $2\rho - 1 = 1 - \alpha n \mu \geq \frac{1}{2}$, so we can get

$$\sum_{k=3}^{K-1}\rho^{K-1-k}r_k^2 \leq 4\rho^{K-3}r_3^2 + 2\sum_{k=3}^{K-1}\rho^{K-1-k}\left(Ac_k + 2b_k + 2a_k n\right)^2.$$

From Lemma 6 we know that

$$r_3^2 \leq 3n^2\varsigma^2 + 27n^2 L_{2,\infty}^2\Delta_2^2 + 12n^2 L_{2,\infty}^2\Delta_1^2,$$

and

$$(Ac_k + 2b_k + 2a_k n)^2 \leq 6A^2\varsigma^2 + 24n^2 L_\infty^2\Delta_k^2 + 54A^2 L_{2,\infty}^2\Delta_{k-1}^2 + 384n^2 L_\infty^2\Delta_{k-1}^2$$
$$+ 24A^2 L_{2,\infty}^2\Delta_{k-2}^2 + 384n^2 L_\infty^2\Delta_{k-2}^2.$$

For the latter part, we can get

$$\sum_{k=3}^{K-1}\rho^{K-1-k}\left(Ac_k + 2b_k + 2a_k n\right)^2$$

$$\leq \frac{6A^2\varsigma^2}{1-\rho} + 24n^2 L_\infty^2 \sum_{k=3}^{K-1}\rho^{K-1-k}\Delta_k^2$$

$$+ 54\rho^{-1}A^2 L_{2,\infty}^2 \sum_{k=3}^{K-1}\rho^{K-1-(k-1)}\Delta_{k-1}^2 + 384\rho^{-1}n^2 L_\infty^2 \sum_{k=3}^{K-1}\rho^{K-1-(k-1)}\Delta_{k-1}^2$$

$$+ 24\rho^{-2}A^2 L_{2,\infty}^2 \sum_{k=3}^{K-1}\rho^{K-1-(k-2)}\Delta_{k-2}^2 + 384\rho^{-2}n^2 L_\infty^2 \sum_{k=3}^{K-1}\rho^{K-1-(k-2)}\Delta_{k-2}^2$$

$$\leq \frac{6A^2\varsigma^2}{1-\rho} + (792\rho^{-2}n^2 L_\infty^2 + 78\rho^{-2}A^2 L_{2,\infty}^2)\sum_{k=3}^{K-1}\rho^{K-1-k}\Delta_k^2$$

$$+ (24\rho^{-2}A^2 L_{2,\infty}^2 + 384\rho^{-2}n^2 L_\infty^2)\rho^{K-2}\Delta_1^2$$

$$+ (78\rho^{-2}A^2 L_{2,\infty}^2 + 768\rho^{-2}n^2 L_\infty^2)\rho^{K-3}\Delta_2^2.$$

Combine everything together,

$$\sum_{k=3}^{K-1}\rho^{K-1-k}r_k^2 \leq 4\rho^{K-3}r_3^2 + 2\sum_{k=3}^{K-1}\rho^{K-1-k}\left(Ac_k + 2b_k + 2a_k n\right)^2$$

$$\leq 12\rho^{K-3}n^2\varsigma^2 + 12n^2 L_{2,\infty}^2\rho^{K-3}\Delta_2^2 + 48\rho^{-1}n^2 L_{2,\infty}^2\rho^{K-2}\Delta_1^2$$

$$+ \frac{12A^2\varsigma^2}{1-\rho} + 2(792\rho^{-2}n^2 L_\infty^2 + 78\rho^{-2}A^2 L_{2,\infty}^2)\sum_{k=3}^{K-1}\rho^{K-1-k}\Delta_k^2$$

$$+ 2(24\rho^{-2}A^2L_{2,\infty}^2 + 384\rho^{-2}n^2L_\infty^2)\rho^{K-2}\Delta_1^2$$
$$+ 2(78\rho^{-2}A^2L_{2,\infty}^2 + 768\rho^{-2}n^2L_\infty^2)\rho^{K-3}\Delta_2^2$$
$$\leq 12\rho^{K-3}n^2\varsigma^2 + \frac{12A^2\varsigma^2}{1-\rho} + 2(792\rho^{-2}n^2L_\infty^2 + 78\rho^{-2}A^2L_{2,\infty}^2)\sum_{k=3}^{K-1}\rho^{K-1-k}\Delta_k^2$$
$$+ 2(24\rho^{-2}A^2L_{2,\infty}^2 + 384\rho^{-2}n^2L_\infty^2 + 24\rho^{-1}n^2L_{2,\infty}^2)\rho^{K-2}\Delta_1^2$$
$$+ 2(78\rho^{-2}A^2L_{2,\infty}^2 + 768\rho^{-2}n^2L_\infty^2 + 6n^2L_{2,\infty}^2)\rho^{K-3}\Delta_2^2.$$

Note that similar to Lemma 4, we get

$$\sum_{k=3}^{K-1}\rho^{K-1-k}\Delta_k^2$$

$$\leq 5\alpha^2\sum_{k=3}^{K-1}\rho^{K-1-k}r_k^2 + 5\alpha^2n^2L_\infty^2\sum_{k=3}^{K-1}\rho^{K-1-k}\Delta_k^2 + 20\alpha^2n^2L_\infty^2\sum_{k=3}^{K-1}\rho^{K-1-k}\Delta_{k-1}^2$$

$$+ 20\alpha^2n^2L_\infty^2\sum_{k=3}^{K-1}\rho^{K-1-k}\Delta_{k-2}^2 + 5\alpha^2n^2\sum_{k=3}^{K-1}\rho^{K-1-k}\|\nabla f(\boldsymbol{w}_{k-1})\|_\infty^2$$

$$\leq 5\alpha^2\sum_{k=3}^{K-1}r_k^2 + 45\alpha^2\rho^{-2}n^2L_\infty^2\sum_{k=3}^{K-1}\rho^{K-1-k}\Delta_k^2 + 40\alpha^2\rho^{-2}n^2L_\infty^2\rho^{K-3}\Delta_2^2$$

$$+ 20\alpha^2\rho^{-2}n^2L_\infty^2\rho^{K-2}\Delta_1^2 + 5\alpha^2n^2\rho^{-1}\sum_{k=2}^{K-1}\rho^{K-1-k}\|\nabla f(\boldsymbol{w}_k)\|_\infty^2$$

$$\leq 60\alpha^2\rho^{K-3}n^2\varsigma^2 + \frac{60\alpha^2A^2\varsigma^2}{1-\rho} + 10\alpha^2(792\rho^{-2}n^2L_\infty^2 + 78\rho^{-2}A^2L_{2,\infty}^2)\sum_{k=3}^{K-1}\rho^{K-1-k}\Delta_k^2$$

$$+ 10\alpha^2(24\rho^{-2}A^2L_{2,\infty}^2 + 384\rho^{-2}n^2L_\infty^2 + 24\rho^{-1}n^2L_{2,\infty}^2)\rho^{K-2}\Delta_1^2$$
$$+ 10\alpha^2(78\rho^{-2}A^2L_{2,\infty}^2 + 768\rho^{-2}n^2L_\infty^2 + 6n^2L_{2,\infty}^2)\rho^{K-3}\Delta_2^2$$

$$+ 45\alpha^2\rho^{-2}n^2L_\infty^2\sum_{k=3}^{K-1}\rho^{K-1-k}\Delta_k^2 + 40\alpha^2\rho^{-2}n^2L_\infty^2\rho^{K-3}\Delta_2^2$$

$$+ 20\alpha^2\rho^{-2}n^2L_\infty^2\rho^{K-2}\Delta_1^2 + 5\alpha^2n^2\rho^{-1}\sum_{k=2}^{K-1}\rho^{K-1-k}\|\nabla f(\boldsymbol{w}_k)\|_\infty^2$$

$$\leq 60\alpha^2\rho^{K-3}n^2\varsigma^2 + \frac{60\alpha^2A^2\varsigma^2}{1-\rho} + \rho^{K-2}\Delta_1^2 + \rho^{K-3}\Delta_2^2 + \frac{1}{2}\sum_{k=3}^{K-1}\rho^{K-1-k}\Delta_k^2$$

$$+ 5\alpha^2n^2\rho^{-1}\sum_{k=2}^{K-1}\rho^{K-1-k}\|\nabla f(\boldsymbol{w}_k)\|_\infty^2.$$

where in the last step, we apply the learning rate requirement that

$$\alpha \leq \min\left\{\frac{1}{n\mu}, \frac{1}{nL}, \frac{1}{52(n+A)L_{2,\infty}}, \frac{1}{520nL_\infty}\right\}.$$

Solve for the LHS we obtain

$$\sum_{k=3}^{K-1}\rho^{K-1-k}\Delta_k^2 \leq 120\alpha^2\rho^{K-3}n^2\varsigma^2 + \frac{120\alpha^2A^2\varsigma^2}{1-\rho} + 2\rho^{K-2}\Delta_1^2 + 2\rho^{K-3}\Delta_2^2$$

$$+ 10\alpha^2n^2\rho^{-1}\sum_{k=2}^{K-1}\rho^{K-1-k}\|\nabla f(\boldsymbol{w}_k)\|_\infty^2.$$

Note that when $k = 1, 2$,

$$\rho^{K-1-k}\Delta_k^2 = 8\rho^{K-1-k}\alpha^2 n^2 \|\nabla f(\boldsymbol{w}_k)\|_\infty^2 + 8\rho^{K-1-k}\alpha^2 n^2 \varsigma^2,$$

Combine everything together we get

$$\sum_{k=1}^{K-1}\rho^{K-1-k}\Delta_k^2 \leq 1024\alpha^2\rho^K n^2\varsigma^2 + \frac{120\alpha^2 A^2\varsigma^2}{1-\rho} + 64\alpha^2 n^2 \sum_{k=1}^{K-1}\rho^{K-1-k}\|\nabla f(\boldsymbol{w}_k)\|_\infty^2.$$

That completes the proof. $\qquad\square$