# OpenReview forum: "GraB: Finding Provably Better Data Permutations than Random Reshuffling"
_NeurIPS.cc/2022/Conference — NeurIPS 2022 Accept_

### Official Review · Reviewer_UoVJ · 2022-07-10

**Rating:** 6
**Confidence:** 4
**Soundness:** 3 good
**Presentation:** 4 excellent
**Contribution:** 4 excellent

**Summary:**

Recent works have shown the possibility of speed up by using good orderings of data samples rather than shuffling them randomly. This paper identifies a mapping between the problem of finding good orderings with "herding" problem leading to a new method that provably improves convergence. The method is further improved to reduce its memory usage leading to a practical algorithm. The algorithm is tested on various tasks and models and is able to yield improved results not just in terms of speed but also in terms of final accuracy.

**Questions:**

* I did not understand how did you arrive at line 491 from 489. In particular, how do you cancel norm_inf(grad) with norm_2(grad) without needing the dimension?

* Since GraB is very well optimized in terms of memory, it seems it should be applicable to larger models such as ResNet. Is there any specific reason that experiments on CIFAR10 were done using LeNet?

* While this can be possibly out of the scope of this paper, I was wondering how much the orderings change throughout training. Especially since the new ordering is based on a re-ordering of the previous epoch.  In particular, does the process converge to a single ordering?

* Would it make sense to run Herding just once at the first epoch to obtain a good ordering?

* How does solving Herding offline compare with GraB? While it is not very practical, is there a huge difference?

**Limitations:**

There is not much discussion around the limitation of the final algorithm, GraB. In particular, Given the low overhead of GraB, I would be interested to know if there are cases that it is still better to use Random Reshuffling? Also, are there cases where GraB can not be currently applied?


**Strengths And Weaknesses:**

This paper provides a practical method for choosing orderings that theoretically converges faster than random reshuffling. This is proven both in theory (when T>n^2). Obtaining these results needs additional smoothness assumptions on infinity norm (in addition to L2) which are justified given that they are guaranteed to some extent (depending on the dimension d) given the L2 smoothness assumption.

Given that the algorithm overhead seems quite low (an additional vector sum at each iteration), this method should be applicable in many cases. Therefore, I believe this to be an important contribution both theoretically and practically and interesting to the community. The experiments are done on different tasks encompassing both image classification and NLP. However, the models used are not the ones that obtain State of the Art results. I believe these results should be included given that the algorithm is being proposed as a replacement for Random Reshuffling and would be happy to update my evaluation after examining them.

Overall, the writing is clear and the paper is well organized so it is easy to follow. I spotted a minor typo:

* In Theorem 3 and Lemma 4, it says Algorithm 6. Do you mean Algorithm 4?

---

> ### Author Response · Authors · 2022-08-02
> **Response to Reviewer UoVJ**
>
> Thanks for giving the excellent reviews! We’ve summarized your questions and answered them separately below. Please let us know if there is any follow-up question.
>
> > I did not understand how you arrived at line 491 from 489. In particular, how do you cancel norm_inf(grad) with norm_2(grad) without needing the dimension?
>
> Thank you for this careful observation! Note that in line 489, the infinite norm can be naturally bounded by the L2 norm on the RHS of the inequality. And so that the norm can be moved to the LHS, by setting the learning rate appropriately, it gives us line 491.
> We have submitted a revised appendix to make this clearer. Please refer to the updated Appendix B line 504 red text.
>
> > Since GraB is very well optimized in terms of memory, it seems it should be applicable to larger models such as ResNet. Is there any specific reason that experiments on CIFAR10 were done using LeNet?
>
> (This question is also raised by Reviewer baYc, and so the answers partially overlap)
>
> Thank you for this great suggestion! Note that in the standard implementation of Resnet-20 on CIFAR-10, the data augmentation technique is used on all the images in every epoch (i.e., random flipping and random cropping, the details can be found in: https://arxiv.org/pdf/1512.03385.pdf), which implies that the actual images used to train are different in each epoch even if the original images are fixed. This makes Resnet-20/CIFAR-10 *not* exactly a finite-sum problem defined in line 20-25. (Lu et al., 2022) discusses how to handle this case via quasi-monte-carlo sampling, but that is beyond the scope of ordering a finite number of data points that we study here.
>
> We would also like to point out that the LSTM and BERT-Tiny models tested in the paper are much larger than ResNet-20: ResNet-20 (0.27M), LSTM (6.9M), BERT-Tiny (4.3M).
>
> > While this can be possibly out of the scope of this paper, I was wondering how much the orderings change throughout training. Especially since the new ordering is based on a re-ordering of the previous epoch. In particular, does the process converge to a single ordering?
>
> Thanks for this incisive question! In practice, we observe the ordering is dynamically changing over the entire training. This is also aligned with the intuition, in that the model weights are randomly initialized and so the gradients at the early stage do not contain too much useful information.
>
> To make this clearer, we conduct an ablation study on fixed orders and include the results in the revised appendix (Please refer to the revised Appendix A red text part and Figure 4).
>
> > Would it make sense to run Herding just once at the first epoch to obtain a good ordering?
>
> Thanks for the excellent question! We found both in theory and in practice that a one-time ordering found by Herding does not work well in later part of the training. This is also aligned with the intuition, in that the model weights are randomly initialized and so the gradients at the early stage do not contain too much useful information about what a good ordering will be later in training. This is also tested empirically in our revised Appendix A.
>
> > How does solving Herding offline compare with GraB? While it is not very practical, is there a huge difference?
>
> Thanks for this great question! Aside from the huge difference in terms of the memory consumption, GraB is much more challenging than offline Herding in that it is entirely online and cannot revisit any gradient. This gives GraB a nice linear complexity compared to the (possibly) quadratic complexity on offline Herding.

---

> > ### Comment · Reviewer_UoVJ · 2022-08-08
> > **Response to Rebuttal**
> >
> > Dear Authors,
> >
> > Thank you very much for your replies.
> >
> > > Thank you for this careful observation! Note that in line 489, the infinite norm can be naturally bounded by the L2 norm on the RHS of the inequality. And so that the norm can be moved to the LHS, by setting the learning rate appropriately, it gives us line 491. We have submitted a revised appendix to make this clearer. Please refer to the updated Appendix B line 504 red text.
> >
> > Thank you for the added clarification. This addresses my concern.
> >
> > > Thank you for this great suggestion! Note that in the standard implementation of Resnet-20 on CIFAR-10, the data augmentation technique is used on all the images in every epoch (i.e., random flipping and random cropping, the details can be found in: https://arxiv.org/pdf/1512.03385.pdf), which implies that the actual images used to train are different in each epoch even if the original images are fixed. This makes Resnet-20/CIFAR-10 not exactly a finite-sum problem defined in line 20-25. (Lu et al., 2022) discusses how to handle this case via quasi-monte-carlo sampling, but that is beyond the scope of ordering a finite number of data points that we study here.
> >
> > While I understand training with data augmentation does not fully correspond to the theory, intuitively it seems possible to me that the algorithm would still work in practice. Even if it turns out that GraB does not work well in this case, it would be an interesting future direction in my opinion.
> >
> > On another note, I think you should specify in the paper (Appendix A) that you do not apply the data augmentation for CIFAR10 + LeNet since it is usually used for experiments on CIFAR10.
> >
> > > We would also like to point out that the LSTM and BERT-Tiny models tested in the paper are much larger than ResNet-20: ResNet-20 (0.27M), LSTM (6.9M), BERT-Tiny (4.3M).
> >
> > I understand that the experiment results include larger models and I agree they can show examples of GraB being applied to to large scale problems. However, as the other reviewers also mentioned, I still think results on ResNet would also be of interest even without augmentation as it is more widely used in computer vision.
> >
> > > To make this clearer, we conduct an ablation study on fixed orders and include the results in the revised appendix (Please refer to the revised Appendix A red text part and Figure 4).
> >
> > Thank you very much for adding these interesting results especially for comparing various cases such as convex vs non-convex and initial and final orderings.
> >
> > > Thanks for this great question! Aside from the huge difference in terms of the memory consumption, GraB is much more challenging than offline Herding in that it is entirely online and cannot revisit any gradient. This gives GraB a nice linear complexity compared to the (possibly) quadratic complexity on offline Herding.
> >
> > Thank you very much for the answer. I agree that the offline version has a huge memory footprint, can be slower and is mostly impractical. However, in my question I was wondering if as a trade-off it can provide better models in terms of accuracy.
> >
> >
> > **Summary**: I have read the author rebuttals and still think the ideas in the paper are novel and the resulting algorithm is interesting to the community. However, I think the paper can be strengthened further by adding the results for more classic settings (e.g. ResNet as the model with or without data augmentation).

---

### Official Review · Reviewer_nHa9 · 2022-07-11

**Rating:** 6
**Confidence:** 4
**Soundness:** 3 good
**Presentation:** 3 good
**Contribution:** 3 good

**Summary:**

This paper studies a herding procedure applied for finding better permutation of dataset for Random Reshuffling algorithm. Authors firstly provide explanations of more illustrative but not practically effective algorithm. Then they provide Gradient Balancing technique, which still has the same properties of herding procedure but it is better in practice. Gradient Balancing requires only $O(d)$ memory and $O(n)$ computations instead of $O(nd)$ and $O(n^2)$ respectively.

This work analyses convergence guarantees of Random Reshuffling with herding procedure for non-convex objectives under L-smoothness  and PL assumptions. Moreover, authors use Bounded Gradient Error assumption. Additionally, authors used different smoothness constants for different norms.

Theoretical analysis shows an improvement from $O(n^{1/3}T^{ 2/3})$ to $O(T^{2/3})$, where $n$ is number of data points. This result means that the new method is $O(n^{1/3})$ times faster.

Usually $n$ is very large in machine learning (deep learning) applications, so even $n^{1/3}$ is significant improvement.

Authors conducted a wide range of experiments with logistic regression and popular deep learning architecture. Plots show practical advantage of new methods compared with other permutation based algorithms.




**Questions:**

1) Is it possible to get results under bounded variance assumption or more general assumption?

2) Can you please provide a table with comparison of results between new method and previous approaches? Also, can you please provide results under the same smoothness assumptions?

3) Is it possible to apply the new approach to find a better permutation to use it for Shuffle Once even with some additional computations?

4) Is it possible in general to apply new technique for convex problems?

5) How can you estimate $f^*$ to calculate alpha parameter?



**Limitations:**

As I mentioned before there are three main limitations:

1) Analysis is made only for non-convex functions, convex and strongly convex analysis is desirable.

2) The current assumptions are reasonable, but not standard. The comparison with previous results under the same assumptions.

4) $f^*$ is used for alpha parameter, but it is usually unknown.

3) Experimental evaluation is not transparent. Additional descriptions of tuned parameters is also needed.

Overall, I believe that this paper has good quality with some limitations. I lean towards to weak accept.

**Strengths And Weaknesses:**

Strengths:

This paper has a good structure and it is well-written. In sections 3-5 the idea of herding and gradient balancing is fully described. This paper has 6 algorithms, which is quite informative. All assumptions are clearly stated and described in remarks.

The idea of finding a better permutation is not new and authors mentioned it in literature review. However, the proposed general framework and efficient implementation is interesting.

The theory seems to be sound. I checked appendix briefly and I did not find problems. However, I could miss something because technical lemmas are long.

This paper has a lot of experiments with different large neural networks showing how this method works in practice.

Weaknesses:

This paper considers only Random Reshuffling scheme and it ignores Shuffle Once version, when permutation is computed only at the beginning of training process. It would be also interesting to see how to choose better permutation and fix it to use the same permutation the whole training process.

This paper does not consider distributed version of random reshuffling (https://arxiv.org/abs/2102.06704, https://arxiv.org/abs/2201.11066, https://arxiv.org/abs/2204.13169) and variance reduced random reshuffling (https://arxiv.org/abs/2104.12112, https://arxiv.org/abs/2104.09342, https://arxiv.org/abs/2111.13322). It would be also interesting to see how proposed technique can be applied to distributed case and variance reduction methods.

This paper considers only non-convex objectives. However, even in current experiments authors consider logistic regression models. Theory for strongly convex and general functions could be appreciable.

Smoothness assumptions for different norms are not standard and if we compare new guarantees with previous results we can see that new bounds depend on dimension $d$. In practical applications $d$ is usually large number and probably large than $d^{1/2}$ can be similar to $n^{1/3}$.

Bounded Gradient Error:
$\text { For any } j \in\{1, \cdots, n\} \text { and any } \boldsymbol{w} \in \mathbb{R}^{d} \text {, }$

$\left\|\nabla f\left(w ; x_{j}\right)-\frac{1}{n} \sum_{s=1}^{n} \nabla f\left(w ; x_{s}\right)\right\|_{2} \leq \varsigma$.

This assumption is stricter than classical bounded variance assumption, since the new bound holds for any index $I$, but not in expectation.

In theory parameter alpha depend on $f^*$ value which is usually unknown.

This paper contains a lot of experiments and plots. However, in supplementary materials I cannot find the source code to understand how parameters were chosen or tuned. In supplementary materials authors provide some details about experimental settings, but it is not enough to recover the results.

---

> ### Author Response · Authors · 2022-08-02
> **Response to Reviewer nHa9**
>
> Thanks for giving the excellent reviews! We’ve summarized your questions and answered them separately below. Please let us know if there is any follow-up question.
>
> > This paper considers only the Random Reshuffling scheme and it ignores the Shuffle Once version, when permutation is computed only at the beginning of the training process. It would be also interesting to see how to choose a better permutation and fix it to use the same permutation the whole training process. Is it possible to apply the new approach to find a better permutation to use it for Shuffle Once even with some additional computations?
>
> (This question is also raised by Reviewer UoVJ, and so the answers partially overlap)
>
> Thanks for this incisive question! In practice, we observe the ordering is dynamically changing over the entire training. This is also aligned with the intuition, in that the model weights are randomly initialized and so the gradients at the early stage do not contain too much useful information.
>
> To make this clearer, we conduct an ablation study on fixed orders and include the results in the revised appendix (Please refer to the revised Appendix A red text part and Figure 4).
>
> > Extended Results on distributed learning, variance reduction, convexity
>
> Thank you for the suggestions on extending GraB results to other settings! We want to politely argue that these extensions are definitely interesting, but should not really be thought of as *weakness* of this paper. There are so many topics in optimization: parallelism, quantization, duality, proximality, acceleration, variance reduction, federated learning, convexity, constraints, data ordering, sampling, adaptive learning rate, etc. All of these topics are interesting and important, but it is obviously impossible to iterate over them in a single conference paper, given we still need quite a lot of space to deliver our main idea. On the other hand, we agree with you that GraB is not yet fully understood in these aspects and will leave these directions as future work.
>
> > Smoothness assumptions for different norms are not standard and if we compare new guarantees with previous results we can see that new bounds depend on dimension d. In practical applications d is usually a large number and probably larger than $d^{1/2}$ can be similar to $n^{1/3}$.
>
> Thank you for the careful observation! Note that the induced norm smooth parameter $L_{2,\infty}$ does not necessarily hide any $d$ factor (see line 136-138). It instead gives us more insights on the dependency of model dimension compared to simply introducing a factor d. For example, if we consider a sparse model (which is common in ML/DL applications), where most of the gradient coordinates are zero, then the old bound (with $d$) would give us a loose bound, while $L_{2,\infty}$ would give us a tiger bound.
>
> > Is it possible to get results under bounded variance assumption or more general assumption?
>
> Thank you for the incisive question! Note that the analysis of GraB/Herding is quite different from traditional work in this field such as with-replacement SGD or random reshuffling. In these baseline works, the algorithms are largely affected by the randomness (in sampling, ordering, etc), and thus assumption on expected norm is critical. In contrast, GraB/Herding are generally close to deterministic algorithms, since we construct these orderings. That is why our final bounds are shown in terms of norms, not expected norms (which is stronger compared to showing expected norms in baseline works).
>
> > Can you please provide a table with comparison of results between the new method and previous approaches? Also, can you please provide results under the same smoothness assumptions?
>
> Thank you for the suggestion of adding a summary table! For your convenience, we’ve included this table in the revised appendix (Table 1) in line 462.
> We also would like to point out here that the assumption of the infinity norm is to be aligned with (Harvey et al., 2014) on herding. It does not really affect the overall proof if we replace the objective (Equation 3) with its L2 norm version. On the other hand, the theoretical improvement of GraB over baseline algorithms is independent of the smoothness assumptions. Since the improvement is shown in terms of $n$ while the smoothness is only assumed on a single function that is invariant to $n$.

---

> > ### Author Response · Authors · 2022-08-02
> > **Response to Reviewer nHa9 (continued)**
> >
> > > How can you estimate $f^*$ to calculate alpha parameter?
> >
> > Thank you for this great question! In practice, instead of computing the learning rate directly from the theory, practitioners usually tune them from a given range, so we wouldn’t need to know $f^*$ in practice at all. On the other hand, in theory showing $f^*$ in the learning rate is a standard practice in the optimization community (see for example https://arxiv.org/pdf/1912.02365.pdf). Alternatively, we can also remove the $f(0)-f^*$ from the learning rate expression, which does not affect the asymptotic convergence rate of the algorithms in terms of $T$ or $n$.

---

> > > ### Comment · Reviewer_nHa9 · 2022-08-06
> > > **Thank you for clarification!**
> > >
> > > Dear authors,
> > >
> > > Thank you for this comment. I understand you point that learning rate can be tuned. However, from theoretical point of view having $f^*$ is a disadvantage even in example you provided above. Fine tuning is very expensive process and a lot of energy and other resources are spent for finding good set of parameters. Theory helps to avoid such expensive search, so having implementable formula for learning rate would be appreciable.

---

> > ### Comment · Reviewer_nHa9 · 2022-08-06
> > **Thank you for your comments!**
> >
> > Dear authors,
> >
> > Thank you for your detailed answer and revised version. Sorry for the late response, I am on vacation with bad Internet connection.
> >
> > I checked all reviews, responses and revised version. I appreciate authors' efforts to make the paper clear.
> >
> > >Thanks for this incisive question! In practice, we observe the ordering is dynamically changing over the entire training. This is also aligned with the intuition, in that the model weights are randomly initialized and so the gradients at the early stage do not contain too much useful information.
> >
> > >To make this clearer, we conduct an ablation study on fixed orders and include the results in the revised appendix (Please refer to the revised Appendix A red text part and Figure 4).
> >
> > Thank you for providing additional plots showing that one permutation is not a good option in practice. I am interested if it is possible to say something from theoretical point of view? Is it possible to prove that Shuffle-Once is worse than Random Reshuffling? As far as I remember this paper shows that they have the same bounds for convex and strongly convex cases, but RR is better than Shuffle Once in non convex case (https://arxiv.org/abs/2006.05988). Or is it possible to prove that 1-step GraB is worse than original GraB? I believe this is interesting and important question and I will be happy to know your thoughts.
> >
> > >Thank you for the careful observation! Note that the induced norm smooth parameter $L_{2,\infty} does not necessarily hide any
> > d factor (see line 136-138). It instead gives us more insights on the dependency of model dimension compared to simply introducing a factor d. For example, if we consider a sparse model (which is common in ML/DL applications), where most of the gradient coordinates are zero, then the old bound (with d) would give us a loose bound, while L_{2,\infty} would give us a tiger bound.
> >
> > I agree that different algorithms and proof techniques require different sets of assumptions. However, it makes results incomparable. Is it possible to somehow provide new results under classical assumptions or provide previous results under new assumptions? This will help to understand the advantage of proposed idea.
> >
> > >Thank you for the incisive question! Note that the analysis of GraB/Herding is quite different from traditional work in this field such as with-replacement SGD or random reshuffling. In these baseline works, the algorithms are largely affected by the randomness (in sampling, ordering, etc), and thus assumption on expected norm is critical. In contrast, GraB/Herding are generally close to deterministic algorithms, since we construct these orderings. That is why our final bounds are shown in terms of norms, not expected norms (which is stronger compared to showing expected norms in baseline works).
> >
> > Thank you for the clarification. In this case it might be useful to add results for deterministic shuffle algorithm in table to have full picture: from deterministic results to results of RR and then to GraB. This will show the whole evolution of permutation-based algorithms.
> >
> > >Thank you for the suggestion of adding a summary table! For your convenience, we’ve included this table in the revised appendix (Table 1) in line 462. We also would like to point out here that the assumption of the infinity norm is to be aligned with (Harvey et al., 2014) on herding. It does not really affect the overall proof if we replace the objective (Equation 3) with its L2 norm version. On the other hand, the theoretical improvement of GraB over baseline algorithms is independent of the smoothness assumptions. Since the improvement is shown in terms of n while the smoothness is only assumed on a single function that is invariant to n.
> >
> > Thank you for clarification! Can you please also add results for deterministic shuffle as I mentioned above?
> >
> >
> > I appreciate the authors' effort and I believe that this is paper is close to acceptance with minor flaws, I decide to keep my score for now.

---

> > > ### Author Response · Authors · 2022-08-08
> > > **Response to Reviewer nHa9**
> > >
> > > We truly appreciate Reviewer nHa9 for giving valuable follow-up questions, even during the vacation. We’ve summarized your questions and answered them separately below. We’ve also submitted a revision to reflect some of your suggestions (details below). Please let us know if any question persists.
> > >
> > > > Thank you for providing additional plots showing that one permutation is not a good option in practice. I am interested if it is possible to say something from a theoretical point of view? Is it possible to prove that Shuffle-Once is worse than Random Reshuffling? As far as I remember this paper shows that they have the same bounds for convex and strongly convex cases, but RR is better than Shuffle Once in non convex case (https://arxiv.org/abs/2006.05988). Or is it possible to prove that 1-step GraB is worse than the original GraB? I believe this is an interesting and important question and I will be happy to know your thoughts.
> > >
> > > Thank you for this great question on the theory! We believe that this question actually touches a critical part on the theoretical motivation of GraB design. We illustrated from the theoretical perspective why reordering is critical in line 169-189. The main takeaway there is that reordering only once does not give us a guaranteed $O(1)$ (or $\tilde{O}(1)$) bound on the herding objective, that is, it does not give the guarantee of the Assumption 4 anymore. In other words, if we just order once, the asymptotic convergence rate of the corresponding 1-step GraB will be the same as SO with a fixed order, since the herding norm in Assumption 4 will generally grow with $k$ like SO. In contrast, the reordering in GraB allows us to make herding bound decrease over time exponentially (we provide proof to a general sequence in Lemma 5 of the appendix). This, we believe, is one of the key designs in GraB, and is motivated from the theory.
> > >
> > > (Note: In your original question, you asked “*is it possible to prove that Shuffle-Once is worse than Random Reshuffling*”, we believe you actually meant “*is it possible to prove that 1-step GraB is worse than the actual GraB*”. Please correct us if this is a misunderstanding. Note that Shuffle-Once, Random Reshuffling, 1-step GraB, and actual GraB introduced in the paper are four different algorithms.)
> > >
> > > > I agree that different algorithms and proof techniques require different sets of assumptions. However, it makes results incomparable. Is it possible to somehow provide new results under classical assumptions or provide previous results under new assumptions? This will help to understand the advantage of the proposed idea.
> > >
> > > Thank you for this great suggestion! In the paper $L_\infty$ and $L_{2,\infty}$ were introduced because the herding objective (Equation (3)) is in the form of the infinity norm. If we just change the herding objective to $\ell_2$ norm, all the $L_\infty$ and $L_{2,\infty}$ in the theory can be replaced with $L_2$ without further modifications (since the triangle inequalities will still hold). That being said, we can simply treat $L_\infty$ and $L_{2,\infty}$ as $L_2$ in all the theories with this updated herding objective. In the original paper, we used the infinity norm for herding objective in order to be consistent with (Harvey et al., 2014) and all the discrepancy theory papers mentioned in the paper.
> > >
> > > We agree with you that seeing how the proposed algorithm GraB adapts under different settings and assumptions in theory will be very interesting, but we hope this does not make us overlook the main contribution of this paper, which is a new methodology of ordering data points beyond random reshuffling. To the best of our knowledge, there is no known work that can both practically (on DNN) and provably outperform random reshuffling, we hope this paper can be a first step to fill this gap.
> > >
> > > > Thank you for the clarification. In this case it might be useful to add results for deterministic shuffle algorithms in the table to have a full picture: from deterministic results to results of RR and then to GraB. This will show the whole evolution of permutation-based algorithms.
> > >
> > > >Thank you for your clarification! Can you please also add results for the deterministic shuffle as I mentioned above?
> > >
> > > Thank you for the excellent suggestion, we have updated Table 2 caption in the revised appendix to clarify this (in red text).

---

### Official Review · Reviewer_baYc · 2022-07-12

**Rating:** 7
**Confidence:** 3
**Soundness:** 3 good
**Presentation:** 4 excellent
**Contribution:** 3 good

**Summary:**

This paper considers the problem of minimizing the sum of a differentiable loss function over a set of examples and
builds on the observation by Lu et al. that convergence can be sped up compared to randomized reshuffling by choosing a permutation that minimizes the average gradient error (defined in equation 2). The authors make the following sequence of contributions:
* Section 3: Minimizing the average gradient error is shown to be equivalent to the herding problem in which the goal is to return an ordering of a sequence of vectors such the maximum deviation of the partial sum from the average is minimized.  The authors show that performing herding with a greedy ordering based on the stale gradients from the previous epoch as proposed by Lu et al. can underperform random reshuffling.  However, if herding is performed properly (i.e. if a constant bound is achieved for the herding objective), this ordering will outperform random reshuffling.
* Section 4: Up to this point, the herding methods proposed require storing a gradient for each example which incurs a large amount of memory overhead.  Vector balancing is presented as an online method which can be used to perform herding.  In short, each vector is assigned a sign as it is viewed in a streaming fashion and then a reordering is performed based on these signs (Algorithm 3).  Applying this procedure multiple times (each of which requires a pass through the gradients) can achieve the desired bound for herding.
* Section 5 + 6: Two remaining challenges are considered: how to center vectors when they are observed in a streaming fashion and the fact that without storing the gradients one can only perform the balancing-reordering routine once per epoch. Solutions are presented and shown to be sufficient to obtain essentially the same convergence rate as proper herding; the full algorithm is named SGD with online gradient balancing (GraB). The advantage of the data order produced by GraB is demonstrated in training for multiple machine learning applications.

Yucheng Lu and Si Yi Meng and Christopher De Sa. "A General Analysis of Example-Selection for Stochastic Gradient Descent" ICLR 2022. https://openreview.net/forum?id=7gWSJrP3opB

**Questions:**

(1) In Theorem 1, the convergence rate is written in terms of the $\ell_2$-norm of the gradient squared averaged over epochs.  Is there a reason this isn't written in terms of the minimum over epochs?

(2) Could you comment on what it would take to extend your results to optimization with a learning rate schedule? Is this easier in the case where you change the learning rate in a step-wise fashion rather than something like cosine decay?

**Limitations:**

I did not see a discussion of limitations of the paper.  I would consider the most relevant discussion to be what common changes to the optimizer (e.g. momentum, learning rate schedule) would be challenging to adapt your theory to?

**Strengths And Weaknesses:**

Strengths:
* The paper makes a solid contribution to an important problem in machine learning.
* The development of an algorithm which can find provably better data permutations without storing all the gradients from the previous epoch enables the scaling of such techniques to large-scale problems.
* The paper is well-written and the flow of ideas is generally easy to follow.

Weaknesses:
* The theory and experiments consider a constant learning rate whereas the majority of modern neural network training utilizes a learning rate schedule with learning rate decay towards the end and possibly learning rate warmup at the beginning.
* I think the experiments would be strengthened by including experiments on CIFAR-10 with a more recent architecture than LeNet, e.g. ResNet-20 which achieves a much higher test accuracy.

Comments:
* For clarity, it might be worth explicitly stating that when connecting the herding objective (eq. 3) and the average gradient error (eq. 2) you are presumably using the relationship $||\mathbf{x}||_{\infty} \leq ||\mathbf{x}||_2$.

I did not thoroughly check the proofs provided in the appendix.

---

> ### Author Response · Authors · 2022-08-02
> **Response to Reviewer baYc**
>
> Thanks for giving the excellent reviews! We’ve summarized your questions and answered them separately below. Please let us know if there is any follow-up question.
>
> > The theory and experiments consider a constant learning rate whereas the majority of modern neural network training utilizes a learning rate schedule with learning rate decay towards the end and possibly learning rate warmup at the beginning. Could you comment on what it would take to extend your results to optimization with a learning rate schedule? Is this easier in the case where you change the learning rate in a stepwise fashion rather than something like cosine decay?
>
> Thank you for this incisive question! As illustrated in line 202-209, one critical design of GraB is that it utilizes the gradients computed from the previous epoch to estimate the gradients at the beginning of the current epoch.  That being said, the benefits margin of GraB over baseline Random Reshuffling would be affected by this approximation error. In other words, the smaller the learning rate, the larger the benefits. Fortunately, in most of the ML/DL applications, the learning rates are usually non-increasing over time, or being small in many epochs (even with cosine). This gives GraB more advantage in practice. On the other hand, in our experiment we test one use case (LSTM) where a large learning rate is used (initial learning rate is 5 as introduced in the Appendix A). As the results suggest, even with this large learning rate, it does not prevent GraB from providing good orderings and speeding up the training.
>
> > I think the experiments would be strengthened by including experiments on CIFAR-10 with a more recent architecture than LeNet, e.g. ResNet-20 which achieves a much higher test accuracy.
>
> Thank you for this great suggestion! Note that in the standard implementation of Resnet-20 on CIFAR-10, the data augmentation technique is used on all the images in every epoch (i.e., random flipping and random cropping, the details can be found in: https://arxiv.org/pdf/1512.03385.pdf), which implies that the actual images used to train are different in each epoch even if the original images are fixed. This makes Resnet-20/CIFAR-10 *not* exactly a finite-sum problem defined in line 20-25. (Lu et al., 2022) discusses how to handle this case via quasi-monte-carlo sampling, but that is beyond the scope of ordering a finite number of data points that we study here.
>
> We would also like to point out that the LSTM and BERT-Tiny models tested in the paper are much larger than ResNet-20: ResNet-20 (0.27M), LSTM (6.9M), BERT-Tiny (4.3M).
>
> > In Theorem 1, the convergence rate is written in terms of the l2-norm of the gradient squared averaged over epochs. Is there a reason this isn't written in terms of the minimum over epochs?
>
> Thank you for pointing this out! We notice both expressions are valid and widely adopted in the optimization community. For reference,
>
> Paper using minimum expression (Theorem 4): https://papers.nips.cc/paper/2020/file/c8cc6e90ccbff44c9cee23611711cdc4-Paper.pdf
>
> Paper using average expression (Equation 4):
> https://arxiv.org/pdf/1705.09056.pdf
>
> In fact, the average expression provided in this paper is *a stronger result*. Note that the minimum expression is implied by the average expression but  not the other way around, since for any non-negative sequence, its minimum will always be upper bounded by its average.

---

> > ### Comment · Reviewer_baYc · 2022-08-07
> > **Response to authors**
> >
> > Thank you to the authors for their detailed response.
> >
> > > One critical design of GraB is that it utilizes the gradients computed from the previous epoch to estimate the gradients at the beginning of the current epoch. That being said, the benefits margin of GraB over baseline Random Reshuffling would be affected by this approximation error. In other words, the smaller the learning rate, the larger the benefits.
> >
> > My understanding of the authors’ point: you are essentially saying if you take smaller steps the gradients saved from the previous iteration will better approximate gradient from the current iteration (because you are making smoothness assumptions) which will result in improved performance.
> >
> > > On the other hand, in our experiment we test one use case (LSTM) where a large learning rate is used (initial learning rate is 5 as introduced in the Appendix A). As the results suggest, even with this large learning rate, it does not prevent GraB from providing good orderings and speeding up the training.
> >
> > Yes, I see now that there is an experiment included that uses a stepped learning rate schedule with improved performance for GraB which is great.  My main question is not about can you use smaller learning rates, but how would you think about a learning rate schedule with respect to your theory?  For example, Theorem 3 gives a rate of convergence assuming that alpha is set to some fixed value.  Is it clear that a similar convergence result holds if I instead use a stepped learning rate schedule or cosine decay, i.e. if the learning rate is changing over time?  If not, what are the challenges in this extension?
> >
> > > Note that in the standard implementation of ResNet-20 on CIFAR-10, the data augmentation technique is used on all the images in every epoch…which implies that the actual images used to train are different in each epoch even if the original images are fixed. This makes Resnet-20/CIFAR-10 not exactly a finite-sum problem defined in line 20-25. (Lu et al., 2022) discusses how to handle this case via quasi-monte-carlo sampling, but that is beyond the scope of ordering a finite number of data points that we study here.
> >
> > Understood, but I think then it is important to add a discussion of this limitation and the interaction of your method with other deep learning techniques like learning rate schedules as discussed above.  Given that your experiments emphasize deep learning applications, it will be important for practitioners looking at using your method to know that it is not currently designed to work with data augmentations as this will often be an unfavorable trade-off for vision applications.  I agree proving results for augmentation is beyond the scope of the problem studied, but if accepted, I strongly encourage you to use the extra space to include these points.
> >
> > > We would also like to point out that the LSTM and BERT-Tiny models tested in the paper are much larger than ResNet-20: ResNet-20 (0.27M), LSTM (6.9M), BERT-Tiny (4.3M).
> >
> > I agree these are excellent experiments and applaud the authors for demonstrating their methods on large-scale deep learning problems.  In particular, they clearly demonstrate the scaling benefit of no longer having to store all the stale gradients.
> >
> > My concern with using LeNet is not that the authors have failed to demonstrate that their method scales, but more that using an architecture that the vision community considers modern will increase the paper’s impact.  Although training without augmentation will result in some decrease in the test accuracy, I think in a final version it would be better to include ResNet results alongside a discussion of the limitation of the method not being designed for augmentation.
> >
> > Two points from looking at Figure 2 again: (1) Why is validation perplexity included for LeNet in Fig. 2b rather than validation accuracy as in the other three experiments?  I see the accuracy is included in appendix Fig. 4 but I don’t understand this inconsistency.  (2) Minor, but in panels A and B the validation accuracy is a percentage (i.e. 67.5) but in panel D it is written as 0.675.  I would recommend making this consistent.
> >
> > > In fact, the average expression provided in this paper is a stronger result. Note that the minimum expression is implied by the average expression but not the other way around, since for any non-negative sequence, its minimum will always be upper bounded by its average.
> >
> > This makes sense and addresses my concern.
> >
> > **Summary:** I have read the other reviews and responses and thank the authors for their efforts to clarify and improve the paper.  I maintain the paper makes a solid contribution to an important problem as reflected by my score.  I encourage the authors to add a discussion of how their theory interacts with common deep learning techniques (learning rate schedule, data augmentation, etc.) to increase the utility of their paper to practitioners.

---

> > > ### Author Response · Authors · 2022-08-08
> > > **Response to Reviewer baYc**
> > >
> > > We want to thank Reviewer baYc for giving such careful read to our paper, and providing incisive follow-up feedback! We’ve summarized your questions and answered them separately below. We’ve also submitted a revision to reflect some of your suggestions (details below). Please let us know if any question persists.
> > >
> > > > Yes, I see now that there is an experiment included that uses a stepped learning rate schedule with improved performance for GraB which is great. My main question is not about can you use smaller learning rates, but how would you think about a learning rate schedule with respect to your theory? For example, Theorem 3 gives a rate of convergence assuming that alpha is set to some fixed value. Is it clear that a similar convergence result holds if I instead use a stepped learning rate schedule or cosine decay, i.e. if the learning rate is changing over time? If not, what are the challenges in this extension?
> > >
> > > Thank you for raising this excellent question, and sorry for misunderstanding your question before.
> > >
> > > We believe this can be answered in two cases: (1) If the learning rate only varies among epochs but remained fixed inside a epoch, it will be straightforward to extend the current theory, since the key lemma of bounding maximum motion in one epoch (Lemma 3, 4) will still hold as the learning rate is fixed there. (2) If the learning rate is changing every step (even in the same epoch), it will incur some challenges. More concretely,  the telescoping on the maximum motion norm (e.g., line 592) will require some constraints on the learning rate such as non-increasing. We will make this clearer in the revision.
> > >
> > > > Understood, but I think then it is important to add a discussion of this limitation and the interaction of your method with other deep learning techniques like learning rate schedules as discussed above. Given that your experiments emphasize deep learning applications, it will be important for practitioners looking at using your method to know that it is not currently designed to work with data augmentations as this will often be an unfavorable trade-off for vision applications. I agree proving results for augmentation is beyond the scope of the problem studied, but if accepted, I strongly encourage you to use the extra space to include these points.
> > >
> > > > My concern with using LeNet is not that the authors have failed to demonstrate that their method scales, but more that using an architecture that the vision community considers modern will increase the paper’s impact. Although training without augmentation will result in some decrease in the test accuracy, I think in a final version it would be better to include ResNet results alongside a discussion of the limitation of the method not being designed for augmentation.
> > >
> > > Thank you for making this important suggestion! We agree with you that this discussion should be included in the paper. Indeed, data augmented dataset, although no longer finite-summed, is still used in many cases. We have added a text (line 480-485 in red) in the revised appendix to clarify this, and will make this clearer in the revision.
> > >
> > > > Two points from looking at Figure 2 again: (1) Why is validation perplexity included for LeNet in Fig. 2b rather than validation accuracy as in the other three experiments? I see the accuracy is included in appendix Fig. 4 but I don’t understand this inconsistency. (2) Minor, but in panels A and B the validation accuracy is a percentage (i.e. 67.5) but in panel D it is written as 0.675. I would recommend making this consistent.
> > >
> > > Thank you for taking such a close look at our plots! For (1) Do you mean Fig. 2c? Since Fig. 2b is still shown in accuracy. The gap here is that 2c (LSTM) is a language modeling task while others are classification tasks, and so we used the perplexity for 2c by convention. (2) This is an acute observation! Thank you for pointing that out and we will make it consistent in the revision.

---

### Author Response · Authors · 2022-08-02
**Rebuttal Summary and Overview of Changes**

We thank all the reviewers for their excellent comments. We summarized each reviewer’s concerns and addressed them separately. Please let us know if there are any remaining questions.

We’ve also submitted a revised version of the appendix to address some concerns from the reviewers. To summarize,

* Page 15 red text includes a new set of ablation study on GraB with fixed orders. The experiment shows insights on how initial fixed order and final fixed order affects the overall convergence, on both convex and non-convex problems. (Reviewer nHa9, UoVJ)
* Table 1 (highlighted in red text) on page 16 summarizes the main theoretical results in this paper. (Reviewer nHa9)
* Page 17 red text clarifies one of the derivation steps between infinity norm and L2 norm. (Reviewer UoVJ)

---

### Meta-Review · Area_Chair_w1NJ · 2022-08-30

**Recommendation:** Accept
**Confidence:** Certain

**Metareview:**

This paper improves the Random Reshuffling method via a herding procedure aimed at finding a better permutation of the training dataset. Authors start by providing intuitive explanations based on a practically ineffective algorithm, and subsequently propose a gradient balancing technique; this enjoys the favorable properties of the herding procedure but is better in practice as it requires $O(n)$ times less memory and computation, where $n$ is number of data points. The authors provide convergence guarantees for Random Reshuffling with empowered with their herding procedure for smooth non-convex problems, under the standard L-smoothness and PL assumptions. Importantly, their theory points to an improvement from $O(n^{1/3}T^{2/3})$ to $O(T^{2/3})$. This result means that the new method is $O(n^{1/3})$ times faster, which is significant.

The reviewers were supportive of the paper, and described its various contributions in the following positive ways:

- Authors conducted a wide range of experiments with logistic regression and popular deep learning architecture. Plots show practical advantage of new methods compared with other permutation based algorithms.
- The paper makes a solid contribution to an important problem in machine learning.
- The development of an algorithm which can find provably better data permutations without storing all the gradients from the previous epoch enables the scaling of such techniques to large-scale problems.
- The paper is well-written and the flow of ideas is generally easy to follow.
- This paper has a good structure and it is well-written.
- This paper studies 6 algorithms, which is quite informative.
- All assumptions are clearly stated and described in remarks.
- The idea of finding a better permutation is not new and authors mentioned it in literature review. However, the proposed general framework and efficient implementation is interesting.
- The theory seems to be sound. I checked appendix briefly and I did not find problems.
- This paper has a lot of experiments with different large neural networks showing how this method works in practice.
- This paper provides a practical method for choosing orderings that theoretically converges faster than random reshuffling.
- Given that the algorithm overhead seems quite low (an additional vector sum at each iteration), this method should be applicable in many cases. Therefore, I believe this to be an important contribution both theoretically and practically and interesting to the community.
- The experiments are done on different tasks encompassing both image classification and NLP.
- Overall, the writing is clear and the paper is well organized so it is easy to follow.

This is a clear acceptance case in my view; the authors were supportive and the rebuttal and the subsequent discussion clarified and addressed most issues. I would request the authors to make sure all criticism will be properly addressed in the camera ready version of the paper.

Congratulations on a nice paper!

AC



**Award:**

No

---

### Decision · Program_Chairs · 2022-09-14

Accept